# $\beta$-DQN: Diverse Exploration via Learning a Behavior Function

## Abstract

Efficient exploration remains a pivotal challenge in reinforcement learning (RL). While numerous methods have been proposed, their lack of simplicity, generality and computational efficiency often lead researchers to choose simple techniques such as $\epsilon$-greedy. Motivated by these considerations, we propose $\beta$-DQN. This method improves exploration by constructing a set of diverse polices through a behavior function $\beta$ learned from the replay memory. First, $\beta$ differentiates actions based on their frequency at each state, which can be used to design strategies for better state coverage. Second, we constrain temporal difference (TD) learning to in-sample data and derive two functions $Q$ and $Q_{mask}$. Function $Q$ may overestimate unseen actions, providing a foundation for bias correction exploration. $Q_{mask}$ reduces the values of unseen actions in $Q$ using $\beta$ as an action mask, thus yields a greedy policy that purely exploit in-sample data. We combine $\beta, Q, Q_{mask}$ to construct a set of policies ranging from exploration to exploitation. Then an adaptive meta-controller selects an effective policy for each episode. $\beta$-DQN is straightforward to implement, imposes minimal hyper-parameter tuning demands, and adds a modest computational overhead to DQN. Our experiments, conducted on simple and challenging exploration domains, demonstrate $\beta$-DQN significantly enhances performance and exhibits broad applicability across a wide range of tasks.

## 1 Introduction

Exploration is considered as a major challenge in deep reinforcement learning (DRL) (Sutton & Barto, 2018; Yang et al., 2021). The agent needs to trade off between exploiting current knowledge for known rewards and exploring the environment for future potential rewards. Despite many complex methods have been proposed for efficient exploration, the most commonly used ones are still simple methods like $\epsilon$-greedy and entropy regularization (Mnih et al., 2015; Schulman et al., 2017). Possible reasons come from two aspects. One is these advanced methods need meticulous hyper-parameters tuning and much computational overhead (Badia et al., 2020a;b; Fan et al., 2023). Another aspect is these methods adopt specialized inductive biases, which may achieve high performance in specific hard exploration environments, but tend to underperform simple methods on a broader range of domains, highlighting the lack of generality (Burda et al., 2019; Taiga et al., 2020).

We improve exploration while taking into account the following considerations: (1) Simplicity. We aim to make clear improvement while also keep the method simple, making it straightforward to implement and has less burden on hyper-parameters tuning. (2) Mild increase on computational overhead. While the primary focus lies in sample efficiency in RL, we aim to strike a balance that avoids substantial increase in training time. Our goal is to develop a method that is both effective and efficient. (3) General for various domains. The method should be general that is applicable for a wide range of tasks, rather than narrowly focusing on some hard exploration games.

In this paper, we design an exploration method by only additionally learning a behavior function $\beta$ from the replay memory using supervised learning. Learned from the current replay memory, $\beta$ can distinguish which actions have been frequently explored and which have not. This can be used to design strategies for better state coverage, for example by taking the action with the least probability at each state. In addition, we use $\beta$ to constrain TD learning to bootstrap from in-sample state-action pairs as shown in Eq. (2). The $Q$ function learns in-sample estimation from the replay memory and generalizes at missing data. The $Q$ function itself may overestimate at unseen state-action pairs, we

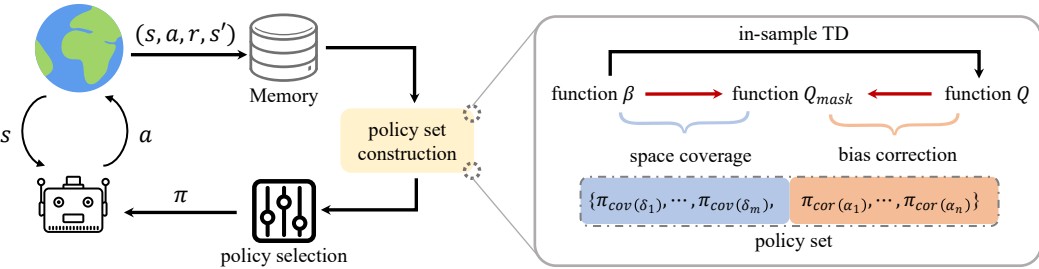

Figure 1: An overview of our method. We learn a behavior function $\beta$ from replay memory, constrain the TD learning to in-sample data to learn function $Q$, and mask $Q$ with $\beta$ to get $Q_{mask}$. With $\beta$, $Q$ and $Q_{mask}$, we construct a set of policies ranging from exploration for space coverage and bias correction to pure exploitation. A meta-controller is designed to adaptively select a policy to interact with then environment for each episode.

use this property to design exploration strategies to try these actions for bias correction (Simmons-Edler et al., 2021; Schaul et al., 2022). Further, if we reduce $Q$ values at unseen actions determined by $\beta$, denoted as $Q_{mask}$, we can derive a greedy policy that purely exploits current experience in the memory. This share the same purpose as offline RL that tries to maximize the cumulative reward limited to a dataset (Lange et al., 2012; Levine et al., 2020), which leads to a conservative policy with best possible performance. The three functions $\beta$, $Q$ and $Q_{mask}$ all have a clear purpose, we use them to construct a set of diverse polices ranging from exploration for space coverage and bias correction to pure exploitation. With the set of diverse policies, we consider the policy selection as a non-stationary multi-armed bandit problem (MAB) (Garivier & Moulines, 2008). We design a meta-controller to adaptively select an effective policy to interact with the environment for each episode, which enables flexibility without an accompanying hyper-parameter-tuning burden.

Our method have several advantages. (1) We only additionally learn a behavior function which is straightforward to implement and computational efficient compared to some other methods (Badia et al., 2020a; Kim et al., 2023). (2) When constructing the policy set, we do not inject inductive biases specialized for one specific task, so the method is general and applicable across a wide range of domains. (3) Our method interleaves exploitation and exploration at intra-episodic level, carries out temporal-extended exploration and is state-dependent, which is considered as the most effective way (Pislar et al., 2022; Dabney et al., 2021). We report promising results on MinAtar (Young & Tian, 2019) and MiniGrid (Chevalier-Boisvert et al., 2023) to show our method significantly enhances performance and exhibits broad applicability in both easy and hard exploration domains.

## 2 RELATED WORK

Reinforcement learning (RL) is generally known as learning by trial and error. If something has not been encountered, it cannot be learned (Pislar et al., 2022). This makes exploration a central challenge for RL. The most commonly used exploration strategies are simple dithering methods like $\epsilon$-greedy and entropy regularization (Mnih et al., 2015; Schulman et al., 2017; Haarnoja et al., 2018). These methods are general but considered inefficient because they are state-independent and lack temporal persistence. Inducing a consistent, state-dependent exploration policy over multiple time steps is the direction that has always been pursued (Osband et al., 2016; Sekar et al., 2020; Ecoffet et al., 2021; Dabney et al., 2021; Simmons-Edler et al., 2021; Pislar et al., 2022).

Bootstrapped DQN (Osband et al., 2016) induces temporally-extended exploration by building up $K$ bootstrapped estimates of the Q-value function in parallel and sample a single $Q$ function for the duration of one episode. The computation increases linearly with the head number $K$. Temporally-extended $\epsilon$-Greedy ($\epsilon z$-greedy) (Dabney et al., 2021) simply repeats the sampled random action for a random duration, whose exploration is still state-independent. Adding exploration bonuses to environment reward by quantifying the novelty of experience is another line of work. Count-based bonuses encourage agents to visit states with a low visit count, and a lot of work has been proposed to estimate counts in high dimension states spaces (Bellemare et al., 2016; Tang et al., 2017; Ostrovski et al., 2017). Another way is based on prediction error such as Intrinsic Curiosity Module (ICM) (Pathak et al., 2017) and Random Network Distillation (RND) (Burda et al., 2019; Badia et al., 2020b). The intuition being that the prediction error will be low on states that are similar to those previously visited and high on newly visited states. These methods emphasize

tackling difficult exploration problems such as MONTEZUMA'S REVENGE. They usually obtain better performance on hard exploration environments, but often underperform simple methods like $\epsilon$-greedy on easy exploration environments (Taiga et al., 2020), which is not general.

Recent promising works tried to handle the problem with population-based methods, which collect samples with diverse behaviors derived from a population of different exploratory policies (Badia et al., 2020a; Fan & Xiao, 2022; Kapturowski et al., 2023; Fan et al., 2023; Kim et al., 2023). They show powerful performance that outperforms the standard human benchmark on all 57 Atari games (Bellemare et al., 2013). These methods maintain a group of actors with independent parameters, build a distributed systems and interact with the environment around billions of frames. Though there has been a performance gain, the computational cost blows up and is unaffordable for most research communities. This has the unfortunate side effect of widening the gap between those with ample access to computational resources, and those without (Ceron & Castro, 2021). Our goal is not to surpass these works, but to absorb strengths from previous work and design an effective method with mild computational resources, which is affordable for most researchers.

## 3 BACKGROUND

**Markov Decision Process (MDP).** Reinforcement learning (RL) (Sutton & Barto, 2018) is a paradigm of agent learning via interaction. It can be modeled as a MDP $\mathcal{M} = (\mathcal{S}, \mathcal{A}, R, P, \rho_0, \gamma)$. $\mathcal{S}$ is the state space, $\mathcal{A}$ is the action space, $P : \mathcal{S} \times \mathcal{A} \times \mathcal{S} \to [0, 1]$ is the environment transition dynamics, $R : \mathcal{S} \times \mathcal{A} \times \mathcal{S} \to \mathbb{R}$ is the reward function, $\rho_0 : \mathcal{S} \to \mathbb{R}$ is the initial state distribution and $\gamma \in (0, 1)$ is the discount factor. The goal of the agent is to learn an optimal policy that maximizes the the expected discounted cumulative rewards $\mathbb{E}[\sum_{t=0}^{\infty} \gamma^t r_t]$.

**Deep Q-Network (DQN).** Q-learning is a classic algorithm to learn the optimal policy. It learns the $Q$ function with Bellman optimality equation (Bellman, 2021), $Q^*(s, a) = \mathbb{E}[r + \gamma \max_{a'} Q^*(s', a')]$. An optimal policy is then derived by taking an action with maximum $Q$ value at each state. DQN (Mnih et al., 2015) scales up Q-learning by using deep neural networks and experience replay (Lin, 1992). It stores transitions in a replay memory and samples batches of that data uniformly to estimate an action-value function $Q_\theta$ with temporal-difference (TD) learning. A target network with parameters $\theta^-$ copies the parameters from $\theta$ only every $C$ steps to stabilize the computation of learning target $y = r + \gamma \max_{a'} Q_{\theta^-}(s', a')$.

## 4 METHOD

Drawing from insights introduced in Section 2, promising exploration strategies should be state-dependent, temporally-extended and consist of a set of diverse policies. And keeping the simplicity and generality in mind, we design an exploration method that is well-performed on a wide range of domains and computational-affordable for our research community. In Section 4.1, we introduce hwo to get the three basic functions, $\beta$ and $Q$ for exploration and $Q_{mask}$ for exploitation. In Section 4.2, by combing the three functions, we interpolate exploration and exploitation at intra-episodic level and get a set of polices ranging from exploration for space coverage or bias correction to pure exploitation. In Section 4.3, we design an adaptive meta-controller to choose the most promising policy to interact with the environment for each episode. Fig. 1 shows an overview of our method.

### 4.1 LEARNING BASIC FUNCTIONS

We construct three basic functions: two for exploration and one for exploitation. The behavior function $\beta$ is used to explore underexplored actions. The $Q$ function is used to explore overestimated actions. The $Q_{mask}$ which reduces unseen state-action values is used to go to the boundary of current experience, which is exploitation. We get three basic functions with clear purposes, and the only extra computation comes from the learning of the behavior function $\beta$ comparing with DQN.

**Behavior Function.** The behavior function $\beta$ is easy to learn, we sample a batch of data $\mathcal{B}$ and train a network using supervised leaning with cross entropy loss:

$$\mathcal{L}_\beta = -\frac{1}{|\mathcal{B}|} \sum_{(s,a)\in\mathcal{B}} \log \beta(s, a). \tag{1}$$

The $\beta$ can differentiate between actions are frequently taken and those are rarely taken at a state. If we choose actions that are less taken, for example $\pi := \arg\min_a \beta(s,a)$, it is a pure exploration policy persuing better state space coverage, without caring about the performance. We use the same data batch to learn $\beta$ and $Q$, thus there is no extra computation from sampling.

**Action Value Function.** Valued-based methods such as DQN estimate an action-value function $Q$ with temporal-difference (TD) learning. We constrain the learning to in-sample state-action pairs:

$$Q(s,a) \leftarrow r + \gamma \max_{a' : \beta(a'|s') > \epsilon} Q(s', a'). \tag{2}$$

The max operator only bootstraps from actions well-supported in the replay memory determined by $\beta(s,a) > \epsilon$, where $\epsilon$ is a small number. Because the data coverage in the replay memory is limited to a tiny subset of the whole environment space, when combing with deep neural networks, Eq. (2) learns an in-sample estimation at existing state-action pairs and generalizes at missing data. It may erroneously overestimate out-of-distribution state-action pairs to have unrealistic values. The greedy policy based on $Q$, i.e. $\pi := \arg\max_a Q(s,a)$, is an optimistic policy that may take erroneously overestimated actions. This can be one kind of exploration that induces corrective feedback to mitigate biased estimation in $Q$ function (Kumar et al., 2020a; Schaul et al., 2022).

Though the $Q$ function learned by in-sample TD learning may overestimate, the behavior function $\beta$ can differentiate actions between frequently and rarely taken. Combining $Q$ with $\beta$, we can reduce the action values at unseen actions and get a conservative estimate:

$$Q_{mask}(s,a) = \begin{cases} Q(s,a), & \text{if } \beta(s,a) > \epsilon \\ \min_{a \in \mathcal{A}} Q(s,a), & \text{otherwise} \end{cases} \tag{3}$$

If we take greedy actions based on $Q_{mask}$, i.e. $\pi := \arg\max_a Q_{mask}(s,a)$, we get a pure exploitation policy. This is similar to offline/batch reinforcement learning (Lange et al., 2012; Levine et al., 2020). The learning goal is to maximize the cumulative reward limited to a static dataset. This returns a conservative policy with currently best possible performance (Kumar et al., 2020b; Kostrikov et al., 2022; Shrestha et al., 2021; Xiao et al., 2023; Zhang et al., 2023).

### 4.2 CONSTRUCTING POLICY SET

With the three basic functions $\beta, Q$ and $Q_{mask}$ with clear purposes, we construct a set of diverse polices that benefits learning. The principle idea is that we can combine two modes of behavior, an *exploration* mode and an *exploitation* mode. For example, $\epsilon$-greedy is the combination of a random policy and a greedy policy. Previous work (Pislar et al., 2022) has shown that intra-episodic exploration, i.e., change the mode of exploitation and exploration in one episode is the most promising diagram. We interpolate between exploration policy and exploitation policy to get a set of policies that explore at some states and exploit at other states within one episode. We construct two kinds of exploration polices. One is to explore rarely taken actions for better state space coverage, the other is to explore overestimated actions and get corrective feedback for bias correction.

**Exploration for Space Coverage.** Because $\beta$ can differentiate between frequently and rarely taken actions, and $Q_{mask}$ is a conservative estimate which denotes pure exploitation. We combine $\beta$ and $Q_{mask}$ to explore for better state space coverage:

$$\pi_{cov(\delta)} = \begin{cases} \arg\max_a Q_{mask}(s,a), & \text{if } \beta(s,a) > \delta \ \forall \ a \in \mathcal{A} \\ \text{DiscreteU}(\{a : \beta(s,a) \le \delta\}), & \text{otherwise} \end{cases} \tag{4}$$

Here, DiscreteU$(\cdot)$ denotes taking a random element from a discrete set. The intuition for $\pi_{cov(\delta)}$ is that if all actions have been tried several times at a state, we follow the exploitation mode to choose actions and reach the boundary of explored area. Otherwise, we choose an action uniformly from undertaken actions which determined by $\delta$. Here $\delta \in [0,1]$ is a parameter, we can set different values to get a group of polices with different exploration degrees. In particular, $\pi_{cov(0)}$ is the pure exploitation policy following $Q_{mask}$, and $\pi_{cov(1)}$ is a random policy.

**Exploration for Bias Correction.** The action value-based algorithms is known to overestimate action values under certain conditions (Van Hasselt et al., 2016; Fujimoto et al., 2018). Accurate value estimation is critical to extract a good policy in DRL. We combine the overestimated $Q$ and the conservative $Q_{mask}$ to try overestimated actions at different states to get corrective feedback:

$$\pi_{cor(\alpha)} = \arg\max_a (\alpha Q(s,a) + (1-\alpha)Q_{mask}(s,a)). \tag{5}$$

The intuition is that we may want to follow the current best actions at some states and explore overestimated actions at some other states. Here $\alpha \in [0, 1]$ is a parameter that we can set different values to get policies with different exploration degree for bias correction. In particular, $\pi_{cor(0)}$ is the pure exploitation policy following $Q_{mask}$, and $\pi_{cor(1)}$ is the optimistic policy following $Q$.

**Constructing Policy Set.** By setting different $\delta$ and $\alpha$ in $\pi_{cov}$ and $\pi_{cor}$, we generate a policy set $\Pi$ that ranging from exploration for better space coverage and bias correction to exploitation:

$$\Pi = \left\{\pi_{cov(\delta_1)}, \cdots, \pi_{cov(\delta_m)}, \pi_{cor(\alpha_1)}, \cdots, \pi_{cor(\alpha_n)}\right\} \tag{6}$$

The policy set does not inject specialized inductive bias thus is general for a wide range of tasks, and the computation will not increase when we add more polices with different $\delta$ and $\alpha$.

### 4.3 META-CONTROLLER FOR POLICY SELECTION

After we have a set of polices, we need to select an effective policy to interact with the environment for each episode. Similar to previous work (Badia et al., 2020a; Fan & Xiao, 2022; Fan et al., 2023; Kim et al., 2023), we consider the policy selection problem as a non-stationary multi-armed bandit (MAB) (Garivier & Moulines, 2008; Lattimore & Szepesvári, 2020) and each policy in the set is an arm. We design a meta-controller to select policies adaptively.

Assume there are $N$ policies in the policy set $\Pi$. For each episode $k \in \mathbb{N}$, a MAB algorithm chooses an arm $A_k = \pi_i$ among the possible arms $\{\pi_0, \cdots, \pi_{N-1}\}$ conditioned on the sequence of previous actions and returns, and receives an episodic return $R_k(A_k) \in \mathbb{R}$. The returns $\{R_k(\pi)\}_{k \geq 0}$ are modeled by a series of random variables whose distributions could change through time during the learning. Our goal is to get a policy $\pi$ that maximizes the return after a given interaction budget $K$.

We use a sliding-window with length $L \in \mathbb{N}^*$ to adapt to the non-stationarity case, i.e. we only care about the recent $L$ results and $L$ is smaller than the interaction budget $K$. The number of times a policy $\pi_i$ has been selected after $k$ episodes for a window of length $L$ is:

$$N_k(\pi_i, L) = \sum_{m=0 \vee k-L}^{k-1} \mathbb{I}(A_m = \pi_i), \tag{7}$$

where $0 \vee k - L$ means $\max(0, k - L)$, and $\mathbb{I}(\cdot)$ is indicator function:

$$\mathbb{I}(U) = \begin{cases} 1, & \text{if } U \text{ is True} \\ 0, & \text{otherwise.} \end{cases} \tag{8}$$

The empirical mean return of $\pi_i$ until episode $k$ is:

$$\mu_k(\pi_i, L) = \frac{1}{N_k(\pi_i, L)} \sum_{m=0 \vee k-L}^{k-1} R_m(\pi_i)\mathbb{I}(A_m = \pi_i). \tag{9}$$

Next we design a bonus $b$ to encourage exploration. Since the greedy policy derived by $Q_{mask}$ is a pure exploitation policy, we consider an action as an exploration action if it is different from the action taken by policy $\pi := \arg\max_a Q_{mask}(s, a)$. For example, if policy $\pi_i$ is selected to interact with the environment at episode $j$, the exploration bonus is computed as:

$$B_j(\pi_i) = \frac{1}{T_j} \sum_{t=0}^{T_j} \mathbb{I}(\pi_i(s_t) = \arg\max_a Q_{mask}(s_t, a)), \tag{10}$$

where $T_j$ denotes the length for episode $j$, $B_j(\pi_i) \in [0, 1]$ is a count-based bonus for exploration. Then the empirical mean exploration bonus of $\pi_i$ until episode $k$ is:

$$b_k(\pi_i, L) = \frac{1}{N_k(\pi_i, L)} \sum_{m=0 \vee k-L}^{k-1} B_m(\pi_i)\mathbb{I}(A_m = \pi_i). \tag{11}$$

Here, the episode $k$ have not been tried and the summation is until $k - 1$. Then for episode $k$, our meta-controller chooses a policy by considering both the environment return and exploration bonus:

$$A_k = \begin{cases} \pi_i, & \text{if } N_k(\pi_i, L) = 0, \\ \arg\max_{\pi_i}(\mu_k(\pi_i, L) + b_k(\pi_i, L)), & \text{otherwise.} \end{cases} \tag{12}$$

In this formula, if a policy has not been selected in the last $L$ episodes, we will prioritize selecting that policy. Otherwise, a policy that explores more often and also gets high returns is preferred. In implementation, we normalize the return into $[0, 1]$ by $(R_k - R_{\min})/(R_{\max} - R_{\min})$, where $R_{\max}$ and $R_{\min}$ are the maximum and minimum return in the sliding window. Thus $\mu_k$ and $b_k$ are at the same magnitude. The pseudo-code of our method is summarized in Appendix A Algorithm 1.

## 5 EXPERIMENTS

In this section, we aim to answer the following questions: (1) Does our method leads to diverse exploration thus benefits learning? (2) Does our method improve performance on both general and sparse reward domains, and also consumes mild computational overhead? (3) What is the role of the two kinds of exploration policies $\pi_{cov}$ and $\pi_{cor}$ in different environments? (4) Is there difference if we learn $Q$ and $Q_{mask}$ with TD learning and in-sample TD learning separately?

### 5.1 A TOY EXAMPLE

We first give a toy example on CliffWalk environment (Sutton & Barto, 2018) to show the policy diversity and the learning process of our method. CliffWalk has 48 states and 4 actions in total as shown in Fig. 2. The goal is to reach state G at the bottom right starting from the bottom left. The reward of reaching G is 1, dropping into the cliff gives -1, otherwise is 0.

For clear illustration of the policy diversity, we design a specific case where there is only one suboptimal trajectory in the replay memory as shown in Fig. 2 top left. The function $\beta$ learned by Eq. (1) will assign probability 1 to the only existing action at these states and other actions as 0. The top right figure shows the estimation error of function $Q$ learned by Eq. (2) at seen and unseen actions. We can find $Q$ will learn accurate esti-

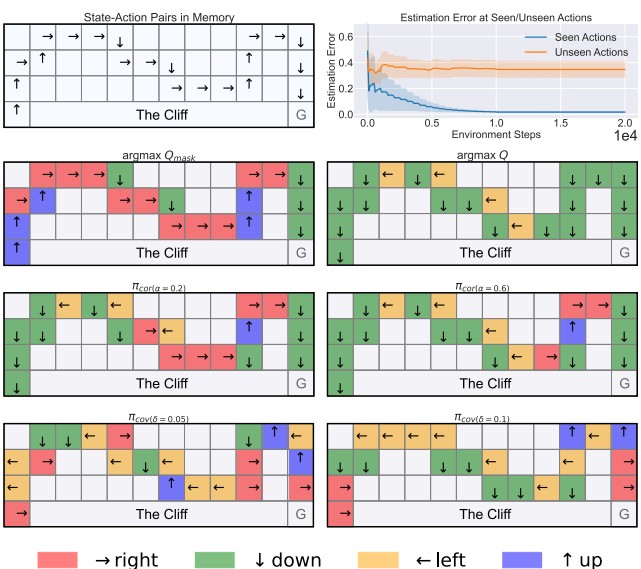

Figure 2: Policy diversity at a specific case. Top left shows the state-action pairs in the memory. Top right shows value errors at seen and unseen actions, implying different roles of $Q$ and $Q_{mask}$. Remaining rows show diverse polices derived from $\beta, Q$ and $Q_{mask}$. These polices take different actions and lead to different states, which benefits the learning.

mates at seen actions but inaccurate estimates at unseen actions, which indicates $Q$ may overestimate at some states and $Q_{mask}$ can yield a best possible policy following current data in the memory. When combining $\beta, Q$ and $Q_{mask}$, we obtain a group of diverse policies as shown in the remaining rows in Fig. 2. These polices take different actions and leading to states that are novel or have biased estimation. Another similar example is given in Appendix C Fig. 10.

In Fig. 3, we show the learning process of our method. The top left figure shows the learning curves of $\beta$-DQN and DQN. Our method finds the goal state and gets the reward 1, while DQN learns how to avoid falling into the cliff but fails to reach the goal state G. The second and third rows show the state coverage of $\beta$-DQN and DQN during learning. Our method explores the whole state space quickly and then maintains the coverage after that. In contrast, DQN agent tries to explore states that are far away from the cliff but neglects to explore the right bottom area that is closed to the cliff and also closed to the goal state. Moreover, the top right figure gives more details of the policy selection in $\beta$-DQN. At the beginning, the exploration policies for data coverage ($\pi_{cov}$) are chosen more frequently. After a good coverage of the whole state space, the exploration policies for bias correction ($\pi_{cor}$) are chosen more frequently in this sparse reward environment.

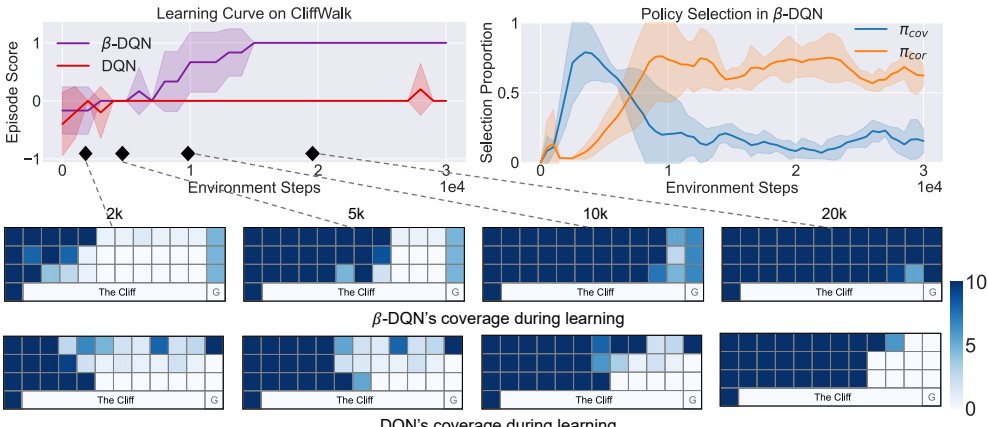

Figure 3: Details during the learning process. The top left shows the learning curves of $\beta$-DQN and DQN. The heatmaps show the state coverage during learning. Our method explores the whole space quickly and finds the path to the goal state which gives reward 1. In contrast, DQN agent learns how to avoid falling into the cliff but fails to reach the goal state G. The top right figure further shows the policy selection in $\beta$-DQN. It first tries to explore the state space with $\pi_{cov}$, and then converts to correct inaccurate estimation using $\pi_{cor}$.

## 5.2 OVERALL PERFORMANCE

We evaluate our method on MiniGrid (Chevalier-Boisvert et al., 2018) and MinAtar (Young & Tian, 2019) based on OpenAI Gym interface (Brockman et al., 2016). MiniGrid implements many tasks in the grid world environment. Most of the games are designed with sparse rewards which is hard to succeed. MinAtar is an image-based miniaturized version of Atari games (Bellemare et al., 2013) which maintains the mechanics of the original games as much as possible and is much faster. For MiniGrid, the map is randomly generated at each episode. For MinAtar, objects are randomly generated at different time steps. So all these environments require the policy to generalize across different configurations. More details about these environments can be found in Appendix B.

We compare our method with DQN (Mnih et al., 2015), bootstrapped DQN (Osband et al., 2016), $\epsilon z$-greedy (Dabney et al., 2021), RND (Burda et al., 2019) and LESSON (Kim et al., 2023). All algorithms are based on DQN but employ different exploration strategies. RND is specialized for sparse reward environments, other methods are general for all kinds of domains. We use the same network architecture for all algorithms as used in MinAtar (Young & Tian, 2019). We search the learning rates for baselines among {3e-4,1e-4, 3e-5} and report the best performance. For our method, though it induces several hyper-parameters, we use one group of parameters across all environments. We fix the policy set $\Pi = \{\pi_{\text{cov}(0.05)}, \pi_{\text{cov}(0.1)}, \pi_{\text{cor}(0)}, \pi_{\text{cor}(0.1)}, \pi_{\text{cor}(0.2)}, \cdots, \pi_{\text{cor}(1)}\}$, and sliding-window length $L = 1000$. Other common parameters are set the same as shown in Appendix A Table 2. We run each experiment with 10 different random seeds. Each run consists of 5 million steps. The performance is evaluated by running 30 episodes after every 100k steps.

Table 1: Overall performance on MiniGrid (success rate) and MinAtar (final score). Numbers in bold represent the method that achieves the best performance. The last row shows the computational overhead compared with DQN. Our method achieves best performance on most games and adds mild computational overhead.

|  | Environment | DQN | BootDQN | $\epsilon z$-greedy | RND | LESSON | $\beta$-DQN (Ours) |
|---|---|---|---|---|---|---|---|
| MiniGrid | DoorKey | 0.44 | 0.11 | 0.0 | **0.99** | 0.86 | 0.98 |
|  | Unlock | 0.22 | 0.17 | 0.0 | 0.95 | 0.64 | **0.99** |
|  | RedBlueDoors | 0.43 | 0.46 | 0.0 | **1.0** | 0.73 | 0.38 |
|  | SimpleCrossing-Easy | **1.0** | **1.0** | 0.95 | 0.95 | 0.97 | 0.99 |
|  | SimpleCrossing-Hard | **1.0** | 0.81 | 0.05 | 0.93 | 0.6 | **1.0** |
|  | LavaCrossing-Easy | 0.29 | 0.66 | 0.26 | 0.68 | 0.75 | **0.84** |
|  | LavaCrossing-Hard | 0.0 | 0.01 | 0.0 | **0.39** | 0.06 | 0.16 |
| MinAtar | Asterix | 22.78 | 22.54 | 18.79 | 13.4 | 18.43 | **39.09** |
|  | Breakout | 16.69 | 21.88 | 19.06 | 14.1 | 17.71 | **29.04** |
|  | Freeway | 60.78 | 59.94 | 59.68 | 49.26 | 54.38 | **62.56** |
|  | Seaquest | 14.66 | 14.31 | 16.98 | 5.61 | 9.41 | **33.23** |
|  | SpaceInvaders | 67.28 | 69.91 | 68.7 | 31.58 | 55.94 | **98.28** |
| Computational Overhead |  | 100% | 195.34 % | 94.32 % | 152.57 % | 371.07 % | 138.78 % |

The final scores are shown in Table 1. We report mean success rate on MiniGrid and mean return on MinAtar. Our method consistently demonstrates superior performance across a wide range of environments, encompassing both general and sparse reward environments. Bootstrapped DQN observes some mild improvement on MinAtar and MiniGrid, indicating its generality but limited performance improvement. $\epsilon z$-greedy finds no obvious improvement on MinAtar and a big drop on MiniGrid. Because repeating the same actions several times blindly may make the agent bump into a wall again and again, which wastes a lot of trials. This indicates even with the injection of temporal persistence, the state-independent exploration is inefficient. RND performs the best on most of the sparse reward environments but performs the worst on MinAtar, which indicates it is not a general method. LESSON improves on MiniGrid to some extend, but performs somehow worse on MinAtar. And it takes much more computational overhead (371%) comparing with DQN than ours (138%) as shown in the last row. $\epsilon z$-greedy run a little faster than DQN because it sample a random action and act for a random duration. This will consume less inference from $Q$ network. To summary, our method is general, effective and computational efficient. Besides the final scores, we show the learning curves for each environment in Appendix C Fig. 11.

## 5.3 ANALYSIS OF OUR METHOD

Our method construct a set of diverse polices with the behavior function $\beta$. Then a meta-controller is used to select an effective policy for each episode. There are some interesting questions we delve deeper: (1) What kind of policy in the policy set is preferred to select by the meta-controller during learning? (2) Which policy performs better, $\arg\max_a Q$ or $\arg\max_a Q_{mask}$? (3) Is there difference if we learn $Q$ and $Q_{mask}$ with TD learning and in-sample TD learning separately? (4) Since we can set different $\delta$ and $\alpha$ to construct the policy set, what is the influence of the policy set size?

### 5.3.1 THE ROLE OF THE TWO KINDS OF EXPLORATION POLICIES

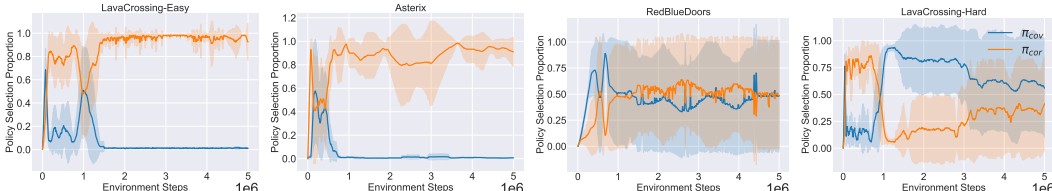

Figure 4: Policy selection during learning. In simple (LavaCrossing-Easy) or dense reward (Asterix) environments, exploring for bias correction plays more important role. In hard exploration environments (RedBlueDoors, LavaCrossing-Hard), the two kinds of polices interleave and result in a more intricate selection pattern.

We construct two kinds of exploration polices in our policy set, $\pi_{cov}$ for state space coverage and $\pi_{cor}$ for bias correction. In Fig. 4, We illustrate the selection proportions of the two kinds of polices within the siding-window during the learning process. Here, $\pi_{cov}$ includes $\{\pi_{\text{cov}(0.05)}, \pi_{\text{cov}(0.1)}\}$, $\pi_{cor}$ includes $\{\pi_{\text{cor}(0.1)}, \pi_{\text{cor}(0.2)}, \cdots, \pi_{\text{cor}(1)}\}$. We can find in simple (LavaCrossing-Easy) or dense reward (Asterix) environments, exploring for bias correction plays more important role. In hard exploration environments such as RedBlueDoors and LavaCrossing-Hard, the two types of policies interleave, resulting in a more intricate selection pattern. This indicates pure exploration itself does not benefit the learning, because novel states may not correlate with improved rewards (Bellemare et al., 2016; Simmons-Edler et al., 2021). In simple environments, it is usually more beneficial to find more low-hanging-fruit rewards, rather than spending much effort exploring novel areas. In hard exploration environments, it is the case that state-novelty exploration plays more important role and find something. This parallels the principles of depth-first search (DFS) and breadth-first search (BFS). When encountering positive rewards, our approach adopts a depth-first exploration, delving deeper into the discovered areas for further exploration. Conversely, in the absence of immediate rewards, we shift towards a breadth-first strategy, exploring widely in search of promising areas.

### 5.3.2 THE PERFORMANCE OF POLICES IN THE POLICY SET

We have three basic functions in our method, we show the performance of them in Fig. 5. The policy $\arg\min_a \beta$ always takes actions with the least probability thus learns nothing. The policy

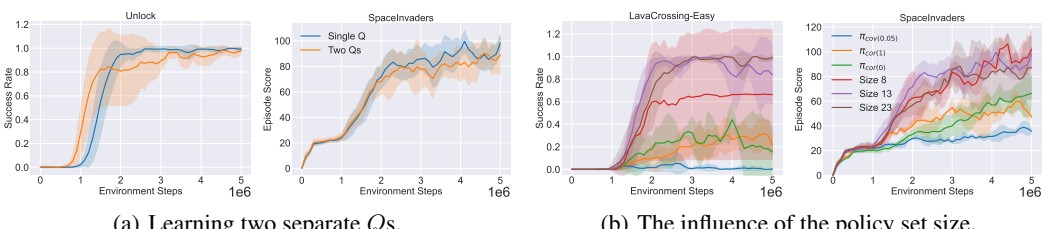

Figure 5: The performance of the three basic polices during learning. $\arg\min_a \beta$ learns nothing since it does not care about performance. $\arg\max_a Q_{mask}$ chooses in-sample greedy actions and performs the best. $\arg\max_a Q$ take greedy actions among the whole action space and may take overestimated actions.

$\arg\max_a Q_{mask}$ chooses greedy actions that is well-supported in the replay memory and performs the best. The policy $\arg\max_a Q$ takes greedy actions among the whole action space, it may take overestimated actions thus the performance is not as stable as $\arg\max_a Q_{mask}$. This result aligns with our expectations, highlighting the distinct purposes of the three basic functions.

### 5.3.3 PERFORMANCE ANALYSIS WITH DIFFERENT CONFIGURATIONS.

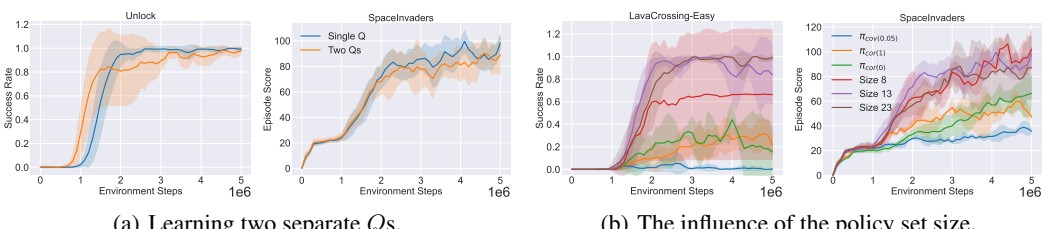

(a) Learning two separate $Q$s.  (b) The influence of the policy set size.

Figure 6: Ablation studies on different configurations. (a) We learn two $Q$s separately and find no obvious difference, which means it is enough to derive two $Q$s from Eq. (2). (b) We construct policy sets with different sizes. There is clear performance improvement when we include all the three functions in the policy set.

**Learning Two Separate $Q$ Functions.** Our method learns one $Q$ function with Eq. (2) and obtain $Q$ and $Q_{mask}$. The intuition is that though Eq. (2) give us a conservative estimate based on in-distribution data, it may still overestimate at unseen state-action pairs. In Fig. 6(a), we show the ablation that learns two separate $Q$ functions with TD learning and in-sample TD learning. We find no obvious difference across environments, which means learning one $Q$ function is enough to get both conservative and optimistic estimation and is more computational efficient.

**Size of Policy Set.** We can construct policy sets with different sizes, and Fig. 6(b) shows the influence. Here, $\pi_{cov(0.05)}, \pi_{cor(0)}, \pi_{cor(1)}$ means there is only one policy. And others show the size of the policy set that combining all three basic functions. More details can be found in Appendix C.3.4. We can find $\pi_{cov(0.05)}$ almost learns nothing, which means only focusing on space coverage does no benefit the leaning. Though $\pi_{cor(0)}$ and $\pi_{cor(1)}$ both learns something, they perform poorer than a big policy set, which means a single policy itself is not enough to get good performance due to the lack of diverse exploration. In contrast, when we combine the three basic functions, we get obvious performance gain with larger set sizes, which emphasize the importance of diverse exploration.

## 6 CONCLUSION

In this paper, we improve exploration by constructing a group of diverse polices via only additionally learning a behavior policy $\beta$ from the replay memory with supervised learning. With $\beta$, we construct a set of exploration policies ranging from exploration for space coverage and bias correction to pure exploitation. Then an adaptive meta-controller is designed to select the most effective policy to interact with the environment for each episode. Our method is simple, general and adds little computational overhead to DQN. Experiments conducted on MinAtar and MiniGrid demonstrate our method is effective and exhibits broad applicability in both easy and hard exploration domains. Future work could be extending our method to environments with continuous action space.

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

## A    IMPLEMENTATION DETAILS

**Hyper-parameters.** All methods are based on DQN. We maintain most parameters the same as DQN and reduce the interaction steps to run more different random seeds. We run each experiment with 5 million steps of interaction with the environment. We proportionally reduce other parameters based on the interaction steps. The $\epsilon$-greedy exploration is linearly decayed from 1 to 0.01 in 1 million steps. The target network is updated every 1000 steps. The replay memory size is set as 100,000. The minibatch size is 32. The replay ratio is 0.25 (Fedus et al., 2020), that is to say the $Q$ function is updated once per four environmental steps. The optimizer for the network is Adam. The discount factor is 0.99. Table 2 shows the detailed hyper-parameters that used for all methods.

Besides the common parameters, there are other parameters that are specific to different methods. For bootstrapped DQN, we follow the parameters setting in the original paper (Osband et al., 2016). We split $K = 10$ separate bootstrap heads after the convolutional layer. And the gradients are normalized by $1/K$. The parameter $p$ in Bernoulli mask $\omega_1, \cdots, \omega_K \sim \text{Ber}(p)$ is set as 1 to save on minibatch passes. When evaluate the performance, we combine all the heads into a single ensemble policy by choosing the action with the most votes across heads. For $\epsilon z$-greedy, to decide the duration of random actions, we use a heavy-tailed distribution zeta distribution $(z(n) \propto n^{-\mu})$ with $\mu = 2$. For RND, we the intrinsic reward scale $\alpha$ is set as 10. In LESSON, the temperature parameter $\tau = 0.02$. The Intrinsic reward coefficient $\alpha$ is set as default value in (Kim et al., 2023). For our method $\beta$-DQN, we use $\beta(s, a) > \epsilon$ as a constraint for the max operator to bootstraps from actions in Eq. (2), we use fixed value $\epsilon = 0.05$. We also fix the policy set $\Pi = \{\pi_{\text{cov}(0.05)}, \pi_{\text{cov}(0.1)}, \pi_{\text{cor}(0)}, \pi_{\text{cor}(0.1)}, \pi_{\text{cor}(0.2)}, \cdots, \pi_{\text{cor}(1)}\}$, and sliding-window length $L = 1000$. We count the same states visited in an episode and avoid visiting the same state-action too much immediately, which is an augment of the behavior function. We search the learning rates for all methods among {3e-3,1e-3,3e-5} and report the best performance. For these baselines, we implement DQN, Bootstrapped DQN, and $\epsilon z$-greedy according to the original papers and refer some awesome public codebases like Tianshou (Weng et al., 2022) [1] and Clearnrl (Huang et al., 2022) [2]. RND and LESSON are based on publicly released code [3].

Table 2: Hyper-parameters of DQN on MiniGrid and MinAtar environments.

| Hyperparameter | Value |
| --- | --- |
| Minibatch size | 32 |
| Replay memory size | 100,000 |
| Target network update frequency | 1,000 |
| Replay ratio | 0.25 |
| Discount factor | 0.99 |
| Optimizer | Adam |
| Initial exploration | 1 |
| Final exploration | 0.01 |
| Exploration decay steps | 1M |
| Total steps in environment | 2M |

**Network Architecture.** We use the same network architecture for all algorithms as used in MinAtar baselines (Young & Tian, 2019). It consists of a convolutional layer, followed by a fully connected layer. The convolutional layer has 16 $3 \times 3$ convolutions with stride 1, the fully connected layer has 128 units. These settings are one quarter of the final convolutional layer and fully connected layer of the network used in DQN (Mnih et al., 2015).

For bootstrapped DQN, We split the network of the final layer into $K = 10$ distinct heads, each one is a fully connected layer with 128 units. RND (Burda et al., 2019) involves two more neural

---

[1] https://github.com/thu-ml/tianshou
[2] https://github.com/vwxyzjn/cleanrl
[3] https://github.com/beanie00/LESSON

networks. One is a fixed randomly initialized neural network which takes an observation to an embedding, and a predictor network trained to predict the embedding output by the fixed randomly initialized neural network. For LESSON (Kim et al., 2023), it involves the prediction networks the same as RND. And the prediction-error maximizing (PEM) intra-policy contains a separate Q-function, which estimates the expected sum of prediction-error intrinsic rewards. Besides, it learns an option selection policy $\{\pi_\Omega\}$ and the terminal functions $\{\beta_\omega\}$, thus has more networks to learn.

**Evaluation.** We run each method on each environment with 10 different random seeds, and show the mean score and standard error with solid line and shaded area in Fig. 11. The performance is evaluated by running 30 episodes after every 100K environmental steps. We use $\epsilon$-greedy exploration at evaluation with $\epsilon = 0.01$ to prevent the agent from being stuck at the same state.

---

**Algorithm 1** $\beta$-DQN

---

1: Initialize replay memory $\mathcal{D}$ with fixed size
2: Initialize functions $\beta, Q, Q_{mask}$, and construct policy set $\Pi$ following Eqs. (4) to (6)
3: **for** episode $k = 0$ **to** $K$ **do**
4:     Select a policy $\pi$ according to Eq. (12)
5:     Initialize the environment $s_0 \leftarrow Env$
6:     **for** environments step $t = 0$ **to** $T$ **do**
7:         Select an action $a_t \sim \pi(\cdot|s_t)$
8:         Execute $a_t$ in $Env$ and get $r_t, s_{t+1}$
9:         Store transition $(s_t, a_t, r_t, s_{t+1})$ in $\mathcal{D}$
10:         Update $\beta$ and $Q$ following Eqs. (1) and (2)
11:     **end for**
12: **end for**

---

## B    Environment Details

### B.1    Cliffworld

Cliffworld is a simple navigation task introduced by Sutton & Barto (2018) as shown in Figure 7. There are 48 states in total which is presented as two-dimensional coordinate axes $x$ and $y$. The size of action space is 4, with left, right, up and down. The agent needs to reach the goal state G at the bottom right starting from the start state S at the bottom left. The reward of reaching the goal is +1, dropped into the cliff gives -1, otherwise is 0. We set the discount factor as 0.9 and the max episode steps as 100. The black line on the figure shows the optimal path.

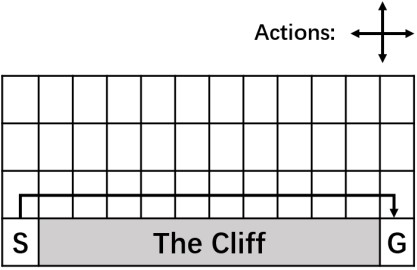

Figure 7: The illustration of Cliffworld environment. Each grid denotes a state, the black line shows the optimal path from start state S to goal state G.

### B.2    MiniGrid

MiniGrid (Chevalier-Boisvert et al., 2018; 2023) [4] is a gridworld Gymnasium (Brockman et al., 2016) environment, which is designed to be particularly simple, lightweight and fast. It implements many tasks in the gridworld environment and most of the games are designed with sparse rewards. We choose seven different tasks as shown in Figure 8.

---

[4] https://github.com/Farama-Foundation/Minigrid

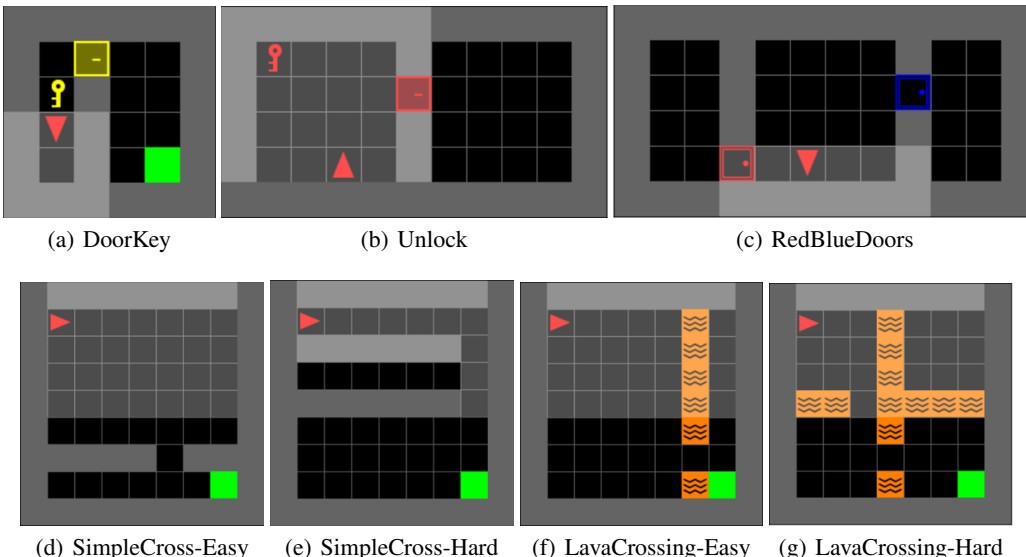

(a) DoorKey      (b) Unlock      (c) RedBlueDoors

(d) SimpleCross-Easy   (e) SimpleCross-Hard   (f) LavaCrossing-Easy   (g) LavaCrossing-Hard

Figure 8: Visualization of MiniGrid environments.

The map for each task is randomly generated at each episode to avoid overfitting to a fixed map. The state is an array with the same size of the map. The red triangle denotes the player, and other objects are denoted with different symbols. The action space is different from tasks. For navigation tasks like SimpleCrossing and LavaCrossing, actions only include turn left, turn right and move forward. For other tasks like DoorKey and Unlock, actions also include pickup a key and open a door. Let MaxSteps be the max episode steps, MapWidth and MapHeight be the width and height of the map. We introduce each task as follows.

**DoorKey.** This task is to first pickup the key, then open the door, and finally reach the goal state (green block). MaxSteps is defined as $10 \times$ MapWidth $\times$ MapHeight. Reaching the goal state will get reward +MaxSteps/100, otherwise there is a penalty reward -0.01 for each step.

**Unlock.** This task is to first pickup the key and then open the door. MaxSteps is defined as $8 \times$ MapHeight$^2$. Opening the door will get reward +MaxSteps/100, otherwise there is a penalty reward -0.01 for each step.

**RedBlueDoors.** This task is to first open the red door and then open the blue door. MaxSteps is defined as $20 \times$ MapHeight$^2$. The agent will get reward +MaxSteps/100 after the red door and the blue door are opened sequentially, otherwise there is a penalty reward -0.01 for each step.

**SimpleCross-Easy/Hard.** This task is to navigate through the room and reach the goal state (green block). Knocking into the wall will keep the agent unmoved. MaxSteps is defined as $4 \times$ MapWidth $\times$ MapHeight. Reaching the goal state will get reward +MaxSteps/100, otherwise there is a penalty reward -0.01 for each step.

**LavaCross-Easy/Hard.** This task is to reach the goal state (green block). Falling into the lava (orange block) will terminate the episode immediately. MaxSteps is defined as $4 \times$ MapWidth $\times$ MapHeight. Reaching the goal state will get reward +MaxSteps/100, falling into the lava will get reward -MaxSteps/100, otherwise there is a penalty reward -0.01 for each step.

### B.3 MINATAR

MinAtar (Young & Tian, 2019) [5] is image-based miniaturized version of Atari environments (Belle-mare et al., 2013), which maintains the mechanics of the original games as much as possible and

---
[5] https://github.com/kenjyoung/MinAtar

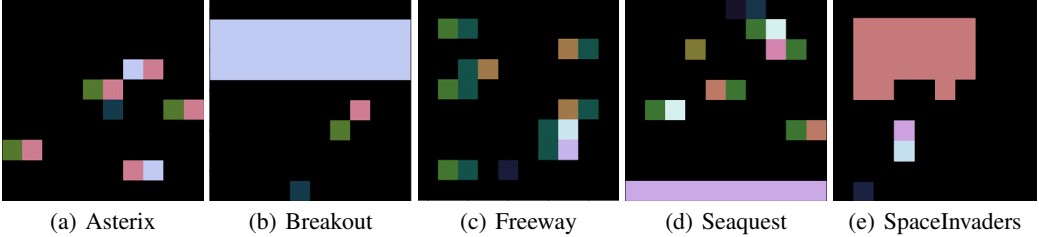

| (a) Asterix | (b) Breakout | (c) Freeway | (d) Seaquest | (e) SpaceInvaders |

Figure 9: Visualization of MinAtar environments.

is much faster than original version. MinAtar implements five Atari games in total, we show the visualization of each game in Figure 9.

**State Space.** Each game provides the agent with a $10 \times 10 \times n$ binary state representation. The $n$ channels correspond to game specific objects, such as ball, paddle and brick in the game Breakout. The objects in each game are randomly generated at different time steps. The difficulty will change as the game progresses, for examples, there will be more objects and the objects will move faster. So these environments needs the policy to generalize across different configurations.

**Action Space.** The action space consists of moving in the 4 cardinal directions, firing, and no-op, and omits diagonal movement as well as actions with simultaneous firing and movement. This simplification increases the difficulty for decision making. In addition, MinAtar games add stochasticity by incorporating sticky-actions, that the environment repeats the last action with probability 0.1 instead of executing the agent's current action. This can avoid deterministic behaviour that simply repeats specific sequences of actions, rather than learning policies that generalize.

**Reward Function.** The rewards in most of the MinAtar environments are either 1 or 0. The only exception is Seaquest, where a bonus reward between 0 and 10 is given proportional to remaining oxygen when surfacing with six divers.

### B.4 WALL-CLOCK TIME COMPARISON

Only reporting sample efficiency cannot tell us how long we need to train an agent for each method. We compare the wall-clock time during training in Table 3. We use *Frames Per Second* (FPS) to measure the training speed. FPS counts the number of frames that the agent interacts with the environment per second.

We test the speed on device with GPU NVIDIA RTX A5000 and CPU AMD EPYC 7313 16-Core Processor. Each time we run 1 experiments and do 3 runs for each method. We can find our method $\beta$-DQN (FPS:627) is slower than DQN (FPS: 870) and faster than methods like Bootstrapped DQN (FPS:445), RND (FPS:570) and LESSON (FPS:234). In addition, since we use a simple network architecture, the computational overhead is relatively low for networks comparing with the computation consumed by environments. If we use a large network, the computation gap will be larger between our method and other methods which have more networks like RND and LESSON. $\epsilon z$-greedy run a little faster than DQN because it sample a random action and act for a random duration. This will consume less inference from $Q$ network.

## C ADDITIONAL EXPERIMENTAL RESULTS

### C.1 TOY EXAMPLE

Besides the example given in Fig. 2, we give another example with full state coverage as shown in Fig. 10. The top left shows all states have been visited. Each state has at least one optimal action in the replay memory, but there are still actions have not been tried. The function $\beta$ learned by Eq. (1) will assign probability uniformly to existing actions at these states and other actions as 0. The top right figure shows the estimation error of function $Q$ learned by Eq. (2) at seen and unseen actions. We can find $Q$ learns accurate estimates at seen actions but inaccurate at unseen actions, which indicates $Q$ may overestimate at some states and $Q_{mask}$ can yield a best possible

Table 3: Wall-clock time comparison between different methods. We use *Frames Per Second* (FPS) to measure the speed of interaction with environments during training. Our method adds mild computational overhead on DQN.

| Method | FPS (mean $\pm$ std) | Computational Overhead |
|---|---|---|
| DQN | $870.64 \pm 7.59$ | 100% |
| Bootstrapped DQN | $445.72 \pm 14.68$ | 195.34% |
| $\epsilon z$-greedy | $923.04 \pm 38.58$ | 94.32% |
| RND | $570.63 \pm 32.31$ | 152.57% |
| LESSON | $234.63 \pm 12.71$ | 371.0% |
| $\beta$-DQN (Ours) | $627.36 \pm 5.34$ | 138.78% |

policy following current data in the memory. A detailed illustration is given at the second row. The blue shading indicates the action values for the greedy actions of $Q_{mask}$ and $Q$. We can find $Q_{mask}$ learn the accurate estimate, but $Q$ overestimates at some state-action pairs and take wrong actions. When combining $\beta, Q$ and $Q_{mask}$, we obtain a group of diverse policies as shown in the remaining rows. These polices take different actions and explore the whole state action space.

## C.2 OVERALL PERFORMANCE

We show the learning curves of each method on MiniGrid and MinAtar in Fig. 11. Each line is the average of running 10 different random seeds. The solid line shows the mean success rate for MiniGrid and the mean return for MinAtar. The shaded area shows the standard error. We can find our method $\beta$-DQN achieves the best performance on most of environments across easy and hard exploration domains, which indicates our method achieves diverse exploration and helps the learning. This highlights that our methods is general and suitable for all kinds of tasks.

## C.3 ADDITIONAL ANALYSIS

### C.3.1 THE ROLE OF THE TWO KINDS OF EXPLORATION POLICIES

We construct two kind of exploration polices in our policy set, $\pi_{cov}$ for state space coverage and $\pi_{cor}$ for bias correction. We can get many of these polices with different $\delta$ and $\alpha$. To show the role of the two kinds of exploration policies, we count them together for clear illustration. For example our main result is based on policy set $\Pi = \{\pi_{\text{cov}(0.05)}, \pi_{\text{cov}(0.1)}, \pi_{\text{cor}(0)}, \pi_{\text{cor}(0.1)}, \pi_{\text{cor}(0.2)}, \cdots, \pi_{\text{cor}(1)}\}$, and sliding-window length $L = 1000$. We count $\{\pi_{\text{cov}(0.05)}, \pi_{\text{cov}(0.1)}\}$ together as $\pi_{cov}$ for state space coverage, and $\{\pi_{\text{cor}(0.1)}, \pi_{\text{cor}(0.2)}, \cdots, \pi_{\text{cor}(1)}\}$ together as $\pi_{cor}$ for bias correction. We also plot $\pi_{\text{cor}(0)}$ separately, since it is a pure exploitation policy.

In Fig. 12, We show the selection proportions of the two kinds of polices in the sliding-window during the learning process. We can find, in all MinAtar environment, the most frequently selected policy are always $\pi_{cor}$. It means following the exploit mode at some states and exploring some overestimated action is enough to get good performance. This indicates, in dense reward environment, there is no need to put much effort to discover hard explored rewards, it is usually more efficient to find more low-hanging-fruit rewards. In MiniGrid environments, the policy selection pattern is more complicated. The state-novelty exploration plays more important role in some environments such as RedBlueDoors and LavaCrossing-Hard. The two types of policies interleave, resulting in a more intricate selection pattern. Our meta-controller parallels the principles of depth-first search (DFS) and breadth-first search (BFS). When encountering positive rewards, our approach adopts a depth-first exploration, delving deeper into the discovered areas for further exploration. Conversely, in the absence of immediate positive feedback, we shift towards a breadth-first strategy, exploring widely in search of promising areas.

### C.3.2 THE PERFORMANCE OF POLICES IN THE POLICY SET

We have three basic functions in our method, one may curious about the performance of the polices derived from the three functions. We show the performance of them in Fig. 13. The policy

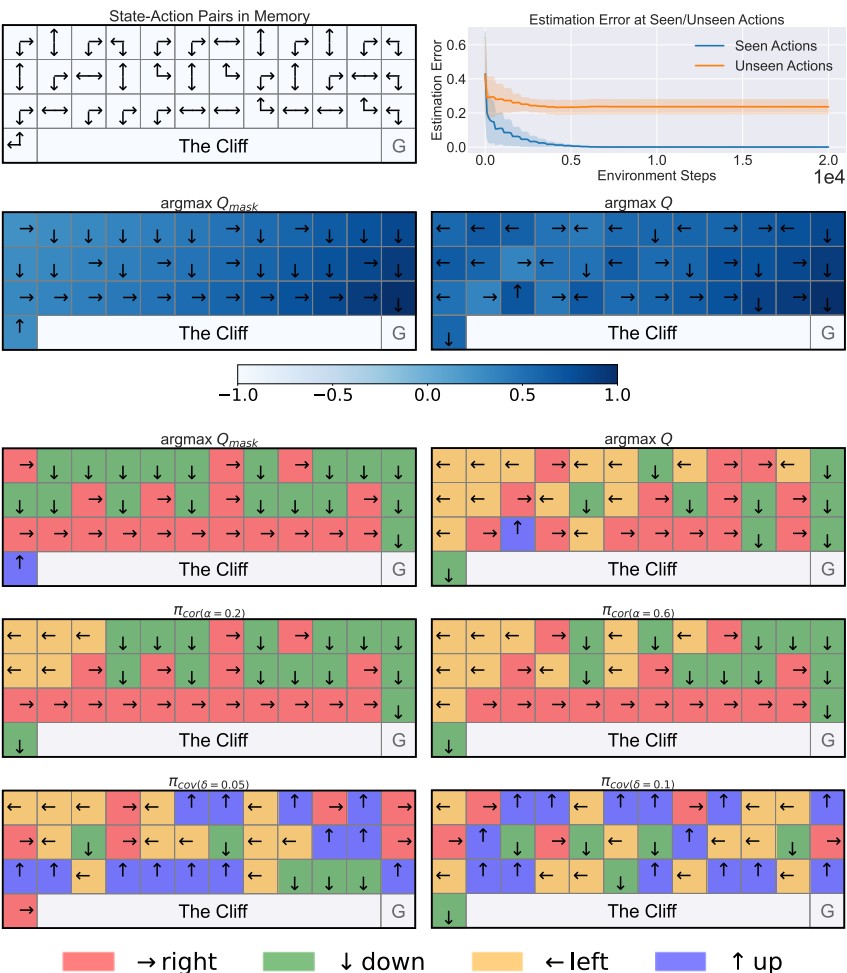

Figure 10: Policy diversity at a specific case. The top left figure shows the state-action pairs in the current memory. All states have been visited and each state has at least one optimal action in the replay memory, but there are still actions have not been tried. The top right figure shows $Q$ value errors at seen and unseen actions. And the second rows explicitly show the action values for the greedy actions of $Q_{mask}$ and $Q$, which indicates $Q$ may overestimate at some states and $Q_{mask}$ can yield a best possible policy, implying different roles of $Q$ and $Q_{mask}$. In the remaining rows, we show diverse polices derived from $\beta, Q$ and $Q_{mask}$. These polices take different actions at these states, which benefits the learning.

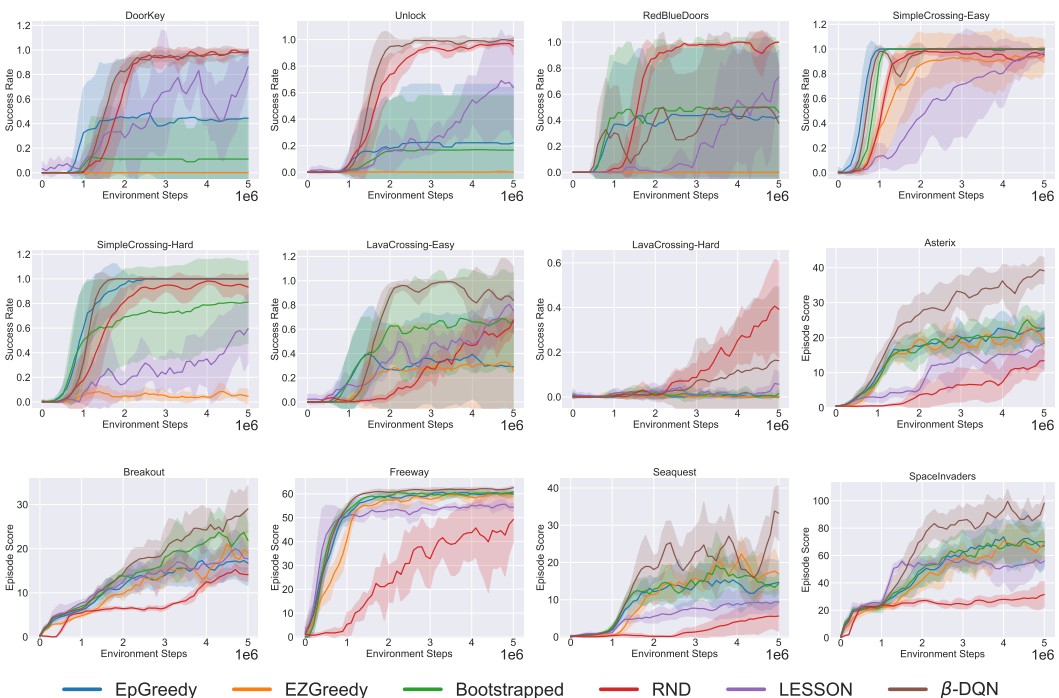

Figure 11: Learning curves on all environments. Our method achieves the best performance on most of environments across easy and hard exploration domains, which indicates our method is general and effective.

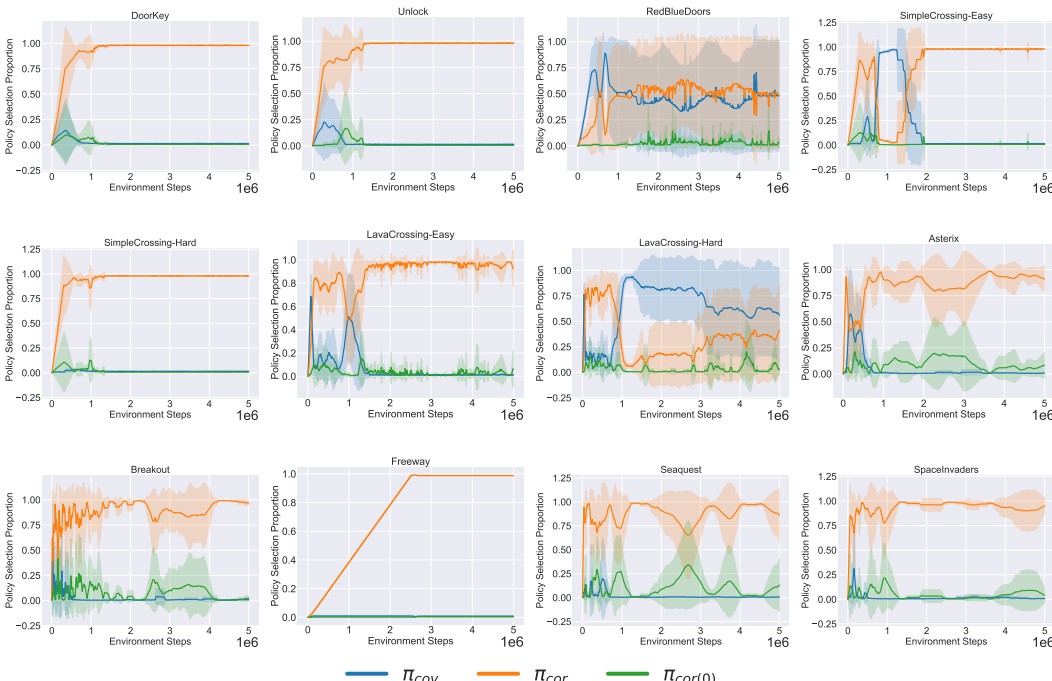

Figure 12: Policy selection proportions during learning on all environments. Policy $\pi_{cor}$ play more important role in simple and dense reward environments to get corrective feedback and correct biased estimation. In hard exploration environments, the two kinds of polices $\pi_{cor}$ and $\pi_{cov}$ interleave and result in a more intricate selection pattern.

$\arg\min_a \beta$ always takes actions with the least probability, it does not care about the performance thus learns nothing. The policy $\arg\max_a Q_{mask}$ chooses greedy actions that is well-supported in the replay memory and performs the best, which is what we expected. The policy $\arg\max_a Q$ takes greedy actions among the whole action space. It may take overestimated actions at some states thus the performance is not as stable as $\arg\max_a Q_{mask}$.

We can also find in some environments like SimpleCrossing-Easy and SimpleCrossing-Hard, $\arg\max_a Q$ performs similar as $\arg\max_a Q_{mask}$. This indicates the space has been fully explored and there is little estimation bias in function $Q$. In contrast, in some other environments like Asterix and SpaceInvaders, there is a large gap between $\arg\max_a Q$ and $\arg\max_a Q_{mask}$. This indicates there is a lot of underexplored overestimated actions waiting for correcting.

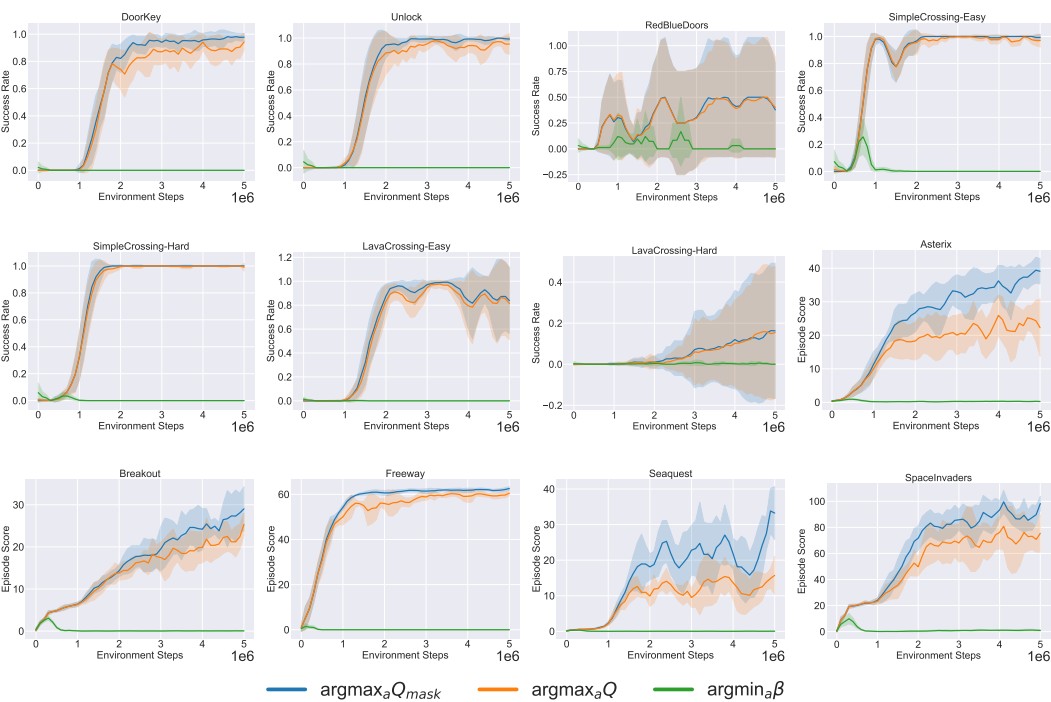

Figure 13: The performance of the three basic polices during learning. $\arg\min_a \beta$ learns nothing since it does not care about performance. $\arg\max_a Q_{mask}$ chooses in-sample greedy actions and performs the best. $\arg\max_a Q$ take greedy actions among the whole action space and may take overestimated actions. Its performance is closed to but a little worse than $\arg\max_a Q_{mask}$.

### C.3.3 LEARNING TWO SEPARATE Q FUNCTIONS

Our method learns one $Q$ function with Eq. (2) and obtain $Q$ and $Q_{mask}$ from the single function. The intuition is that though Eq. (2) gives us a conservative estimate based on in-distribution data, it may still overestimate at unseen state-action pairs as shown in the toy example Figs. 2 and 10. A more natural way is that we can maintain the update rule in DQN unchanged as:

$$Q(s, a) \leftarrow r(s, a) + \gamma \max_{a'} Q(s', a'). \tag{13}$$

And additionally learn another $Q$ following Eq. (2). In this way, we learn $Q$ function with TD learning and $Q_{mask}$ with in-sample TD learning. We show the ablation in Fig. 14. On most of these environment, we find no big difference, which means learning one $Q$ function with Eq. (2) is enough to get both conservative and optimistic estimation and adds less computational overhead.

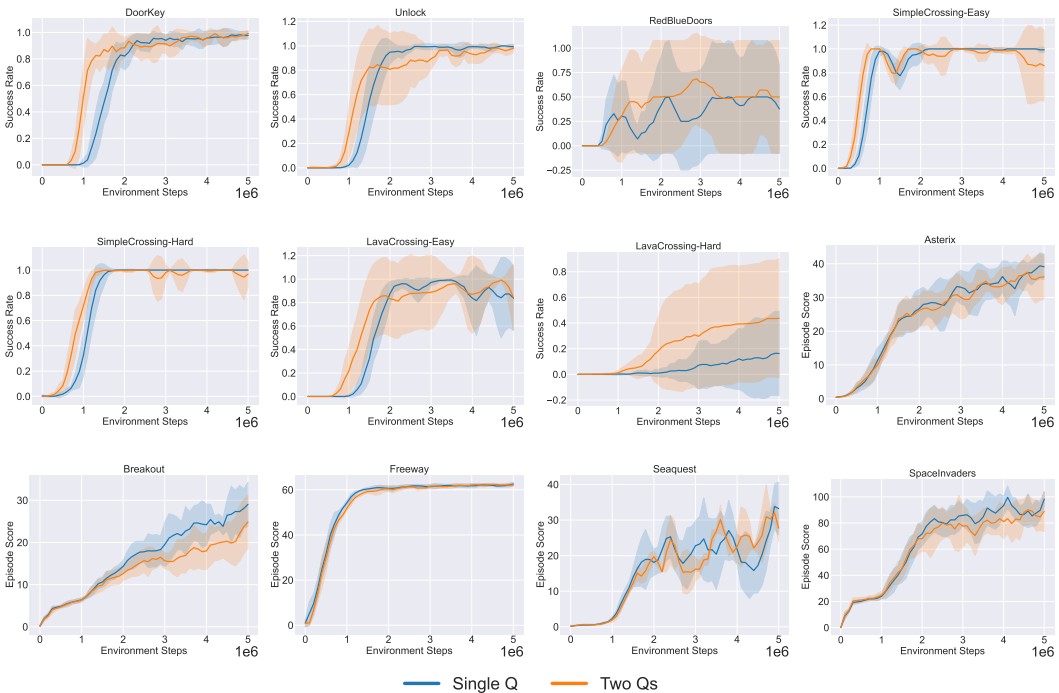

Figure 14: Ablation study on learning two separate $Q$ functions. We learn the $Q$ function with TD learning, and learning $Q_{mask}$ with constrained TD learning and behavior function $\beta$. There is no obvious difference between the two methods, which means learning one $Q$ with constrained TD learning is enough to derive two $Q$ functions.

### C.3.4 THE INFLUENCE OF POLICY SET SIZE.

One benefit of our method is that we can construct policy sets with different sizes without increasing computational overhead. By adding different $\delta$ and $\alpha$, we get larger policy set. We construct different sizes of policy sets as shown in Fig. 15.

The $\pi_{cov(0.05)}, \pi_{cor(0)}, \pi_{cor(1)}$ in the figure means there is only one policy. And others show the size of the policy set that combining all three basic functions like Eq. (6). **Size 8** denotes the policy set $\Pi = \{\pi_{cov(0.05)}, \pi_{cov(0.1)}, \pi_{cor(0)}, \pi_{cor(0.2)}, \pi_{cor(0.4)}, \cdots, \pi_{cor(1)}\}$. **Size 13** denotes the policy set $\Pi = \{\pi_{cov(0.05)}, \pi_{cov(0.1)}, \pi_{cor(0)}, \pi_{cor(0.1)}, \pi_{cor(0.2)}, \cdots, \pi_{cor(1)}\}$, which we used in our main results. **Size 23** denote the policy set $\Pi = \{\pi_{cov(0.05)}, \pi_{cov(0.1)}, \pi_{cor(0)}, \pi_{cor(0.05)}, \pi_{cor(0.1)}, \cdots, \pi_{cor(1)}\}$.

We can find $\pi_{cov(0.05)}$ does not learn anything in most of environments, which indicates only focusing on space coverage does no benefit the leaning. This may because novel states may not correlate with improved rewards (Bellemare et al., 2016; Simmons-Edler et al., 2021). Though $\pi_{cor(0)}$ and $\pi_{cor(1)}$ both learns something, they perform poorer than a big policy set, which indicates a single policy itself is not enough to get good performance due to the lack of diverse exploration. In contrast, when we combine the three of the basic functions, we get obvious performance gain with larger set sizes, which emphasize the importance of diverse exploration. And when we increase the policy size to 13 and 23, there is no big difference within 5 million environmental steps, which may indicate the diversity is similar in this two policy sets.

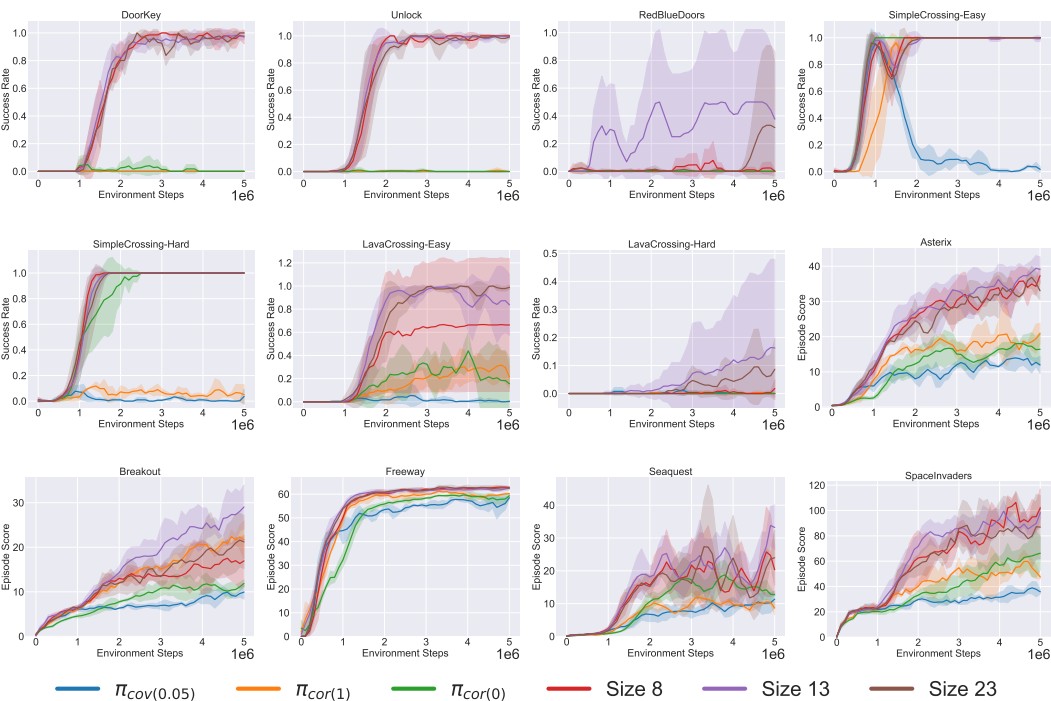

Figure 15: The influence of the policy set size. We construct the policy set with different sizes. The performance improves with increasing policy size and obtain obvious improvement if we contain all the three basic functions in the policy set.

