# OpenReview forum: "$\beta$-DQN: Diverse Exploration via Learning a Behavior Function"
_ICLR.cc/2024/Conference — Submitted to ICLR 2024_

### Official Review · Reviewer_Swey · 2023-10-17

**Soundness:** 3 good
**Presentation:** 3 good
**Contribution:** 2 fair
**Rating:** 6
**Confidence:** 3

**Summary:**

This paper considers the problem of designing exploration schemes that are simple, general, and computationally efficient. Towards this end, the authors propose to combine three basic functions: a new beta function the authors learned to explore under-explored actions, the usual Q function for exploring overestimated actions, and finally a Q_mask function designed for exploitation. A diverse set of policies is constructed from them to encourage space coverage or bias correction, and the specific policy used per episode is selected by a meta-controller in a non-stationary multi-armed bandit fashion. Experiments are performed on a few environments and various components of the proposed method are analyzed, which validates the soundness of the proposed exploration scheme.

**Strengths:**

The authors propose an interesting method to improve exploration. The method is intuitive and the presentation is very clear. It makes sense to me by combining the three basic functions to construct a set of policies that explore for space coverage and another set of policies that explore for bias correction. The toy example on CliffWalk and the related figures are very helpful in understanding/visualizing the method. There are also experimental ablation studies on the method. Overall, the evaluation seems to confirm that this is a sound approach.

**Weaknesses:**

The experiments can be done more comprehensively. As of this manuscript, the method is only evaluated on a selected set of environments. Given the simplicity and generality of the method, it makes sense to conduct a larger-scale experiments, e.g., divide all environments on Atari into easy/hard exploration tasks and see if one can systematically analyze the performance gain/loss of the method and whether interesting insights could be drawn as a whole. While the method works right now as of the current presentation, it's unclear how those environments are selected.

**Questions:**

1. The toy example is presented well with interesting observations. For example, this makes intuitive sense: "At the beginning, the exploration policies for data coverage (πcov) are chosen more frequently. After a good coverage of the whole state space, the exploration policies for bias correction (πcor) are chosen more frequently in this sparse reward environment." The pattern however does not hold and becomes more complicated in Section 5.3.1 (Figure 4). Could the authors perform more experiments on a diverse environments to see if any insights can be obtained collectively wrt the environment charactristics?
2. In Section 5.3.2, "The policy argmax Q takes greedy actions among the whole action space, it may take overestimated actions thus the performance is not as stable as argmax Qmask."  Figure 5 doesn't seem to suggest any clearly difference in terms of stability, though argmax Q_mask does perform better.

---

> ### Author Response · Authors · 2023-11-15
> **Response to Reviewer Swey**
>
> - The pattern however does not hold and becomes more complicated in Section 5.3.1 (Figure 4).
>
> In our current experiments, we find that in simple (LavaCrossing-Easy) or dense reward (Asterix) environments, exploring for bias correction plays a more important role. In hard exploration environments (RedBlue-Doors, LavaCrossing-Hard), the two kinds of policies interleave and result in a more intricate selection pattern.
> We distill these insights as follows: In simpler environments, the emphasis is on efficiently discovering low-hanging-fruit rewards, prioritizing exploitation over extensive exploration. In contrast, in challenging exploration environments, the importance of state-novelty exploration is heightened, resembling the principles of depth-first search (DFS) and breadth-first search (BFS). Our approach adapts to this context: upon encountering positive rewards, it adopts a depth-first exploration strategy, delving deeper into discovered areas for further exploration. Conversely, in the absence of immediate rewards, the strategy shifts towards a breadth-first approach, exploring widely to uncover promising areas.
>
> - argmax Q is not as stable as argmax Qmask in Figure 5.
>
> Thanks for pointing out this. A more precise description would be argmax Q does not perform as well as argmax Qmask because it may select overestimated actions. We will revise the description to accurately reflect this distinction.

---

### Official Review · Reviewer_fvxy · 2023-10-29

**Soundness:** 3 good
**Presentation:** 2 fair
**Contribution:** 2 fair
**Rating:** 5
**Confidence:** 3

**Summary:**

This paper works on improving the exploration-exploitation trade-off in discrete control.
The key idea is to introduce a behaviour network that is trained by supervised update to the replay data.
And further use such a behaviour network to control the behaviour for exploration or exploitation.
The authors further introduce a meta controller to select from a set of potential combinations.

**Strengths:**

The idea is straightforward and easy to follow. The problem addressed in this paper is important.

The authors conducted plenty of empirical studies, and the results seem to be highly replicable.

**Weaknesses:**

please see the questions.

**Questions:**

1. High-level idea: the key insight behind exploration is not simply to visit the actions that are infrequently visited, but to add frequency measure (like the count-based methods) to the value, to perform an exploration-exploitation trade-off. The idea of learning a behaviour density estimator is interesting, but integrating this idea with a count-based MAB exploration algorithm makes the algorithm no longer elegant, and hard to identify the performance improvement.

2. The presentation of the paper can be improved. There are many misleading terminology usages. e.g., the *space coverage*, *bias correction*, what does those terms mean in this context?

3. Is the MAB trained concurrently with the Q networks? How does the choice of hyper-parameters affect learning stability (it was partially revealed in Figure 6). In practice, is there a golden standard for selecting those hyper-parameters?

4. It would be helpful to have a standard deviation in results (table 1.)

5. In Equation (1), it’s not clear what \beta exactly is and how it is parameterised and optimised. Is it a supervised / behaviour cloning policy?

6. Some related works are missing:

Ensemble:
- https://arxiv.org/pdf/1611.01929.pdf discussed ensemble methods in DQN.
- https://arxiv.org/pdf/2209.07288.pdf discussed the ensemble methods using different hyperparameter settings.

Curiosity-Driven Exploration:
- https://arxiv.org/abs/2206.08332 BYOL-Explore is one of the state-of-the-art exploration algorithms for discrete and continuous control
- https://arxiv.org/abs/2211.10515
- https://openreview.net/forum?id=_ptUyYP19mP further improves the RND

7. The authors mentioned Montezuma's Revenge environment in the introduction but did not experiment in this challenging setting. I wonder if the authors could provide results in such a setting.

---

I'm happy to re-evaluate the work if those questions can be addressed.

---

> ### Author Response · Authors · 2023-11-15
> **Response to Reviewer fvxy**
>
> - integrating this idea with a count-based MAB exploration algorithm makes the algorithm no longer elegant.
>
> Integrating our idea with a count-based Multi-Armed Bandit (MAB) exploration algorithm is a plausible approach. Notably, there exist various methods that learn a population of policies and leverage a MAB meta-controller for policy selection [1,2,3].
>
> - The presentation of the paper can be improved.
>
> We appreciate the feedback and will refine the presentation to make the definitions clearer.
>
> - Is the MAB trained concurrently with the Q networks
>
> MAB does not require training. Instead, it performs statistical analysis on past data and continuously updates its statistics as more data is collected. This unique characteristic sets the MAB simple and efficient, allowing it to adapt and refine its policy selection strategy dynamically with the accumulation of experience.
>
> - standard deviation in results (table 1.)
>
> We will add it in the next version.
>
> - what \beta exactly is
>
> $\beta$ serves as an estimation of the frequency of state-action pairs within the replay memory. This estimation is akin to count-based methods, but $\beta$ does not rely on an explicit count. Instead, it operates as a supervised policy, dynamically adapting to the changing environment by providing a probabilistic measure of the likelihood of selecting different state-action pairs.
>
> - related work
>
> We appreciate your suggestion. We will discuss this related work in the next version.
>
>
> [1] Badia, Adrià Puigdomènech, Bilal Piot, Steven Kapturowski, Pablo Sprechmann, Alex Vitvitskyi, Zhaohan Daniel Guo, and Charles Blundell. "Agent57: Outperforming the atari human benchmark." In International conference on machine learning, pp. 507-517. PMLR, 2020.
>
> [2] Fan, Jiajun, and Changnan Xiao. "Generalized Data Distribution Iteration." In International Conference on Machine Learning, pp. 6103-6184. PMLR, 2022.
>
> [3] Fan, Jiajun, Yuzheng Zhuang, Yuecheng Liu, H. A. O. Jianye, Bin Wang, Jiangcheng Zhu, Hao Wang, and Shu-Tao Xia. "Learnable Behavior Control: Breaking Atari Human World Records via Sample-Efficient Behavior Selection." In The Eleventh International Conference on Learning Representations. 2023.

---

### Official Review · Reviewer_J55j · 2023-11-01

**Soundness:** 3 good
**Presentation:** 3 good
**Contribution:** 1 poor
**Rating:** 3
**Confidence:** 4

**Summary:**

$\beta$-DQN is a method proposed to address the challenge of efficient exploration in reinforcement learning. It improves exploration by constructing a set of diverse policies through a behavior function $\beta$ learned from the replay memory. The method is straightforward to implement, imposes minimal hyper-parameter tuning demands, and adds a modest computational overhead to DQN. Experimental results demonstrate that $\beta$-DQN significantly enhances performance and exhibits broad applicability across a wide range of tasks.

**Strengths:**

1. The method proposed in this paper only requires learning a behavior function, which is straightforward to implement and computationally efficient compared to other methods.
2. The method incorporates both exploitation and exploration at the intra-episodic level, allowing for effective temporal-extended exploration that is state-dependent.
3. The paper reports promising results on MinAtar and MiniGrid, demonstrating that the method significantly enhances performance and exhibits broad applicability in both easy and hard exploration domains.

**Weaknesses:**

1. The main concern is that the proposed method has a large overlap with a prior work: BAC [1], but there are neither discussion nor comparison in this paper. For example, the idea of learning in-sample state-action pairs in Eq. (2) is similar to Eq. (4.1) in BAC. The idea of making trade-off between standard and conservative Q-value functions in Eq. (5) is similar to Eq. (4.3) in BAC. Apart from being overlapped with BAC, the idea of constructing a policy set has been extensively studied in literature, e.g. [2].
2. There is a lack of baseline algorithms to compare with. The curiosity-driven exploration algorithms should be included as baselines.
3. The behavior function $\beta$ is learned with supervised learning, and is not robust when facing policy or dynamics shifts.
4. The DQN-style formulation constrains the action space to be discrete. Therefore, the porposed algorithm cannot fit to continuous control tasks.




[1] Ji T, Luo Y, Sun F, et al. Seizing Serendipity: Exploiting the Value of Past Success in Off-Policy Actor-Critic. arXiv preprint arXiv:2306.02865, 2023.

[2] Sun H, Peng Z, Dai B, et al. Novel policy seeking with constrained optimization. arXiv preprint arXiv:2005.10696, 2020.

**Questions:**

See weaknesses.

---

> ### Author Response · Authors · 2023-11-15
> **Response to Reviewer J55j**
>
> - Related work and baselines
>
> We will discuss these related work and add more baselines in the next version.
> BAC [1] is a concurrent work which is also submitted to ICLR2024 (https://openreview.net/forum?id=VPx3Jw2MSk). We will compare with it in the next paper version. While both works consider in-distribution learning, which makes them similar at the high level, the methods used are very different.  BAC blends two operators to trade off exploitation (IQL-style) and exploration (SAC-style), while our method uses a behavior function $\beta$ as a mask to switch the mode between exploitation and exploration. In this way we can explicitly generate a group of policies ranging from pure exploitation to exploration with Q and $\beta$, while BAC cannot.
> Further, the BAC method tunes parameters for each task individually to get high performance, while we used a single set of parameters for all tasks. This reduces the burden of hyper-parameter tuning.
>
> The idea of constructing a policy set has been extensively studied in the literature [2,3,4,5]. All these methods maintain a group of policies with independent parameters. The computational cost blows up and is unaffordable for most research communities. In contrast, our method constructs a set of diverse policies from only one additionally learned behavior function, which is
> straightforward to implement and computationally efficient.
>
> - $\beta$ is learned with supervised learning, and is not robust when facing policy or dynamics shifts.
>
> In online learning, since the replay memory changes with every environment step, $\beta$ and $Q$ keep changing accordingly. The role of $\beta$ is to estimate the current replay memory content, and it should be updated with the changing memory. The supervised learning of $\beta$ is more straightforward and stable compared to the concurrent deep reinforcement learning (DRL) of $Q$. The inherent stability of learning $\beta$ stems from its simplified nature, contributing to a robust training process.
>
> - The DQN-style formulation constrains the action space to be discrete.
>
> The DQN-style formulation, by design, restricts the action space to be discrete. Our method is an improvement for DQN, it cannot be applied directly to continuous action space. Addressing this limitation and extending our method to accommodate continuous action spaces is an intriguing avenue for future work.
>
> [1] Ji T, Luo Y, Sun F, et al. Seizing Serendipity: Exploiting the Value of Past Success in Off-Policy Actor-Critic. arXiv preprint arXiv:2306.02865, 2023.
>
> [2] Sun H, Peng Z, Dai B, et al. Novel policy seeking with constrained optimization. arXiv preprint arXiv:2005.10696, 2020.
>
> [3] Badia, Adrià Puigdomènech, Bilal Piot, Steven Kapturowski, Pablo Sprechmann, Alex Vitvitskyi, Zhaohan Daniel Guo, and Charles Blundell. "Agent57: Outperforming the atari human benchmark." In International conference on machine learning, pp. 507-517. PMLR, 2020.
>
> [4] Fan, Jiajun, and Changnan Xiao. "Generalized Data Distribution Iteration." In International Conference on Machine Learning, pp. 6103-6184. PMLR, 2022.
>
> [5] Fan, Jiajun, Yuzheng Zhuang, Yuecheng Liu, H. A. O. Jianye, Bin Wang, Jiangcheng Zhu, Hao Wang, and Shu-Tao Xia. "Learnable Behavior Control: Breaking Atari Human World Records via Sample-Efficient Behavior Selection." In The Eleventh International Conference on Learning Representations. 2023.

---

### Official Review · Reviewer_G7o1 · 2023-11-04

**Soundness:** 2 fair
**Presentation:** 3 good
**Contribution:** 3 good
**Rating:** 3
**Confidence:** 4

**Summary:**

The authors propose a novel off-policy RL algorithm $\beta$-DQN based on DQN. $\beta$-DQN aims to reduce the computation complexity, and the hyperparameter sensitivity compared to the previous exploration RL algorithms. The authors provide experiments on MinAtar and MiniGrid for back up their claim.

**Strengths:**

The authors propose a novel framework for solving exploration problems in reinforcement learning. The strength can be decomposed as the following:
* Unlike most of the exploration algorithms, the proposed algorithm $\beta$-DQN does not use auxiliary intrinsic reward neural networks to estimate the uncertainty, instead, $\beta$-DQN constructs a set of policy, and treat the RL problem as a MAB (Multi-Armed Bandit) problem by selecting different policies for each episode, with the usage of UCB-alike bonus for exploration.
* The proposed algorithm is much more computationally efficient compared to previous intrinsic-reward based exploration algorithms and has a more structured exploration strategy than \epsilon-greedy.
* $\beta$-DQN adopts the technique of top-$p$ sampling, with a diverse range of $p$ to construct a diverse set of policies with different preferences.

**Weaknesses:**

1. The proposed algorithm is limited to the discrete action domain while most other exploration algorithms do not have such a limitation, and there is no obvious way to directly apply it to the continuous action domains.
2. The environments of choice are not standard, and potentially too easy to solve. The scale of both environments is too small, and it is skeptical that the proposed algorithm does not scale to a more standard discrete action domain like Atari. The details of my concern are the following:
* First, the behavior policy \beta is changing every environment step, which means that the "mask" is changing every step. Given that the experiments are only conducted in environments with a limited diversity of observations, the stability of the algorithm when learning in a more complex environment is concerning.
3. The baselines are not tuned, and the selection of the baseline does not align with the main motivation of the experiment. I am going to decompose my reasons as the following:
* The purpose of this paper is to propose a computationally efficient, hyperparameter-insensitive, and well-performing exploration algorithm. However, $\beta$-DQN has two hyperparameters, $L$ and policy set size. Policy set size controls the hardness of the MAB algorithm, whereas the $L$ affects the process of the learning of MAB. The authors provide an ablation study on the analysis of the sensitivity of the size of the policy set but do not provide the ablation study on $L$, which is not enough to convince the reader that $\beta$-DQN is not sensitive to hyperparameters.
* As the authors mentioned in the paper, most exploration algorithms are sensitive to hyperparameters. For intrinsic-reward based ones, they are particularly sensitive to the bonus scale factor $\alpha$. Specifically, the implementation of RND is from the LESSON paper by Kim et al., but the authors use a different $\alpha$ in this paper and achieved better performance compared to the RND in LESSON paper, which further shows that the the hyperparameter selection can be crucial for the performance of exploration algorithm like RND. Without the systematic tuning of hyperparameters for the baselines, it is hard to convince the readers that $\beta$-DQN is preferred. Additionally, the ablation study of the size of the policy set does not show an insensitivity of $\beta$-DQN to this hyperparameter.
Overall, the experiment setup, including the choice of environments, the hyperparameter selections for baselines, and the ablation study on hyperparameters of $\beta$-DQN, does not provide enough information for the reader to decide whether or not $\beta$-DQN is more hyperparameter insensitive, or performs better than other exploration methods.

**Questions:**

Question:
1. LESSON provides a set of hyperparameters for the MiniGrid domain for RND-based DQN, why do the authors use $\alpha=10$, which is way larger than the hyperparameters that LESSON paper used?

Suggestions:
The authors propose a novel algorithm, that smartly integrates the MAB algorithm in the RL setting to help reduce the computation complexity, however, the claims of "hyperparameter-insensitivity" and "effective exploration" are not sufficiently backed up by the experiment. The detailed suggestions are the following:
1. Tune the hyperparameters of RND, Bootstrapped DQN in a systematic manner, and show that even the best hyperparameter would not yield better performance compared to $\beta$-DQN, i.e. showing that $\beta$-DQN is indeed favorable compared to them.
2. Compare $\beta$-DQN against the public implementation of RND-PPO and Bootstrapped DQN with the original selection of hyperparameters on the Atari domain, on two sets of environments one contains environments with sparse reward and requiring heavy exploration, the other contains environments with dense reward where even $\epsilon$-greedy can perform well.
3. Provide a more systematic ablation study for the hyperparameter of $\beta$-DQN.

---

> ### Author Response · Authors · 2023-11-15
> **Response to Reviewer G7o1**
>
> - limited to the discrete action domain.
>
> Yes. Currently our method is an improvement for DQN, and cannot be applied directly to continuous action space. We will extend it to continuous action space in future work.
>
> - The behavior policy \beta is changing every environment step.
>
> Indeed, the dynamic nature of the behavior policy $\beta$ aligns with our objective. In the context of online learning, the replay memory changes with every environment step. $\beta$ serves as an estimation of the replay memory and should be updated along with the evolving memory content. Supervised learning of $\beta$ is more straightforward than concurrent deep reinforcement learning (DRL) of $Q$, where both $\beta$ and $Q$ are subject to continuous adjustments. The simpler learning task for $\beta$ leads to inherent greater stability and contributes to a robust training process.
>
> - The baselines are not tuned.
>
> We did tune each baseline and achieved higher performance than in the original paper. For example, untuned RND and LESSON by Kim et al. achieve success rates of around 0.1 and 0.2 on LavaCrossingS9N1 (LavaCrossing-Easy in our paper). In our paper, these baseline methods are tuned to achieve 0.63 and 0.75, much higher than the original paper. We found that these intrinsic reward based methods are sensitive to hyperparameters.
>
> - Ablation study and systematic hyperparameters tuning of baselines.
>
> We appreciate the systematic suggestions provided. In our next version, we will conduct an ablation study and undertake a more comprehensive systematic hyperparameter tuning of baselines.
>
> Regarding the ablation study on $L$, a rough search among $L$ in [100, 500, 1000, 2000] on SimpleCrossing-Easy and Asterix found no obvious differences between 500, 1000 and 2000, and lower performance for $L=100$. Based on this preliminary experiment, we fixed $L= 1000$ for all environments. We can do a grid search for $L$ for all environments.

---

> > ### Comment · Reviewer_G7o1 · 2023-11-21
> >
> > Thank you for the clarification. However, the baselines and the environments of choice are still concerning (despite they are better than the results shown in the LESSON paper). I decided to keep my score this time.

---

### Official Review · Reviewer_7r6n · 2023-11-08

**Soundness:** 3 good
**Presentation:** 3 good
**Contribution:** 2 fair
**Rating:** 3
**Confidence:** 4

**Summary:**

This paper proposed a new method to perform diverse exploration in the reinforcement learning setting. A behavior function beta is learned on in-sample data to model the coverage of different state action pairs. Then standard DQN learning learns a state-action value function Q. Based on the behavior function and the state-action value function Q, the authors proposed to construct a new Q_mask function that corresponds to a pure exploitation policy due to that less explored actions are suppressed by altering its Q value. Next the paper constructs two basic policies pi_cov and pi_cor that aim at better coverage and bias correction respectively. From there, a set of diverse policies are created. These diverse policies have different levels of exploration and exploitation that intuitively help with the learning process. A meta controller is designed based on non-stationary multi-armed bandit (MAB) algorithms to select a policy from the diverse set, which is then used to interact with the underlying environment to collect new data. Experiments on synthetic data demonstrate the effectiveness of the proposed methodoloy.

**Strengths:**

(1) The paper studied diverse exploration, which is an important topic in RL.

(2) The proposed method makes intuitive sense, and the way to construct diverse policy set based on interpolation between pi_cov and pi_cor is a reasonable approach.

(3) There are experiments that demonstrate the effectiveness of the proposed method.

**Weaknesses:**

(1) The paper does not have enough technical contribution. The way to construct diverse policies is elementary. It only concerns learning some function beta that models the data coverage and then doing some policy combination. The way to construct meta controller borrows the MAB framework and does not contain enough novel investigations.

(2) The paper has no theoretical results. The proposed method is not built based on foundational theory but instead some intuitions. This makes the methodology not very trustworthy.

(3) The experiments are performed only on synthetic data and very simple environments. It remains not clear how well the method generalizes to more complicated scenarios such as continuous environment.

**Questions:**

(1) The experiments are performed on simple environment with finite state and action space. How does the proposed method behave on more complicated tasks that involve continuous states and actions.

(2) How can we derive theoretical results to understand better why the proposed exploration strategy works well?

---

> ### Author Response · Authors · 2023-11-15
> **Response to Reviewer 7r6n**
>
> - Technical contribution.
>
> Our paper is motivated by the fact that while many complex methods have been proposed for efficient exploration, the most commonly used ones are still simple methods such as $\epsilon$-greedy and entropy regularization [1,2]. Previous literature attributes this to two key challenges of other methods: their increased complexity and lack of generality [3,4]. In response, our contribution is a simple yet effective exploration method which enhances the performance of DQN. Our idea is simple and straightforward to implement, which we consider an advantage.
>
> - No theoretical results.
>
> Our paper is experimental, focused on empirical results rather than theoretical analysis. Other well-known simple yet effective practical exploration approaches include $\epsilon z$-greedy [4] and RND [5]. Our work builds on and extends this empirical tradition, and contributes to the understanding of exploration strategies through a series of experiments.
>
> - Not clear how well .. generalizes to more complicated scenarios such as continuous environments.
>
> As an improvement for DQN, our method does not apply directly to continuous action spaces. However, it is suitable for state spaces that are continuous/high-dimensional. Our current experiments are all on image-based domains. While MinAtar is simpler than Atari it is also high-dimensional, in 4 out of 5 games (all except Breakout) trajectories do not overlap in practice [6]. Even for minigrid, the tasks are not simple for current algorithms. For example, in our results for LavaCrossing-Hard, no algorithm can learn to reliably reach a goal state within 5 million steps. This is still a very challenging testbed worth deeper research.
>
> [1] Mnih, Volodymyr, Koray Kavukcuoglu, David Silver, Andrei A. Rusu, Joel Veness, Marc G. Bellemare, Alex Graves et al. "Human-level control through deep reinforcement learning." nature 518, no. 7540 (2015): 529-533.
>
> [2] Schulman, John, Filip Wolski, Prafulla Dhariwal, Alec Radford, and Oleg Klimov. "Proximal policy optimization algorithms." arXiv preprint arXiv:1707.06347 (2017).
>
> [3] Taiga, Adrien Ali, William Fedus, Marlos C. Machado, Aaron Courville, and Marc G. Bellemare. "On Bonus Based Exploration Methods In The Arcade Learning Environment." In International Conference on Learning Representations. 2020.
>
> [4] Dabney, Will, Georg Ostrovski, and Andre Barreto. "Temporally-Extended ε-Greedy Exploration." In International Conference on Learning Representations. 2021.
>
> [5] Burda, Yuri, Harrison Edwards, Amos Storkey, and Oleg Klimov. "Exploration by random network distillation." In International Conference on Learning Representations. 2019.
>
> [6] Zhang, Hongming, Chenjun Xiao, Han Wang, Jun Jin, and Martin Müller. "Replay Memory as An Empirical MDP: Combining Conservative Estimation with Experience Replay." In The Eleventh International Conference on Learning Representations. 2023.

---

### Meta-Review · Area_Chair_ymBt · 2023-12-09

**Metareview:**

The paper proposes $\beta$-DQN, a method for efficient exploration in reinforcement learning (RL) that constructs diverse policies through a behavior function learned from replay memory. The method aims to maintain simplicity, generality, and computational efficiency. It focuses on enhancing exploration and performance in various domains, mainly using image-based environments.

## Strengths:
- The proposed method addresses the pivotal challenge of exploration in RL with an innovative approach.
- $\beta$-DQN is straightforward to implement, demands minimal hyper-parameter tuning, and adds only modest computational overhead to DQN.
- The paper provides empirical evidence showing the method’s effectiveness across various tasks, demonstrating its broad applicability.

## Weaknesses:
- The technical novelty of the method is questioned, with some reviewers considering its approach elementary.
- There is a lack of theoretical foundation for the proposed method, relying heavily on empirical results.
- The experiments are limited to simpler, synthetic environments, raising questions about the method’s applicability to more complex, real-world scenarios.
- Comparisons with existing methods are insufficient, especially in areas like curiosity-driven exploration algorithms.

## Reviewer Concerns:
- Questions about the method’s scalability and effectiveness in continuous environments.
- Need for more rigorous theoretical analysis to understand why the proposed strategy works.
- Similarities with existing works are not adequately addressed, especially with concurrent works like BAC.

## Author Responses:
- Authors defend the simplicity and empirical focus of their method, comparing it to other simple yet effective exploration methods.
- They acknowledge the method’s current limitation to discrete action spaces, suggesting future work for continuous environments.
- Authors agree to enhance comparisons with existing works and provide more comprehensive experiments in future revisions.

**Justification For Why Not Higher Score:**

While the paper introduces an interesting approach to solving exploration challenges in reinforcement learning, it falls short in technical novelty, theoretical grounding, and comprehensive experimentation. The method’s effectiveness in more complex scenarios remains uncertain. However, its simplicity and initial empirical results are promising. The paper needs significant improvements in terms of addressing the technical depth, providing theoretical insights, and expanding experimental validation to be considered for acceptance in future conferences.

**Justification For Why Not Lower Score:**

N/A

---

### Decision · Program_Chairs · 2024-01-16

Reject